# A Survey on Computer-Aided Diagnosis of Brain Disorders through MRI Based on Machine Learning and Data Mining Methodologies with an Emphasis on Alzheimer Disease Diagnosis and the Contribution of the Multimodal Fusion

**Lilia Lazli** [1,2,3,*] , **Mounir Boukadoum** [2] **and Otmane Ait Mohamed** [4]

1    Department of Electrical engineering, École de Technologie Supérieure (ÉTS), University of Quebec, Montreal, QC H3C 1K3, Canada

2    CoFaMic research Center, Computer Science department, Université du Québec à Montréal (UQAM), University of Quebec, Montreal, QC H3C 3P8, Canada; boukadoum.mounir@uqam.ca

3    Artificial intelligence research group (ERIA), Computer Science laboratory (LRI), Computer Science department, University of Badji Mokhtar Annaba (UBMA), BP. 12 Annaba 23000, Algeria

4    Department of Electrical and Computer Engineering, Concordia University, Montreal, QC H3G 1M8, Canada; otmane.aitmohamed@concordia.ca

*    Correspondence: lilia.lazli.1@ens.etsmtl.ca

**Abstract:** Computer-aided diagnostic (CAD) systems use machine learning methods that provide a synergistic effect between the neuroradiologist and the computer, enabling an efficient and rapid diagnosis of the patient's condition. As part of the early diagnosis of Alzheimer's disease (AD), which is a major public health problem, the CAD system provides a neuropsychological assessment that helps mitigate its effects. The use of data fusion techniques by CAD systems has proven to be useful, they allow for the merging of information relating to the brain and its tissues from MRI, with that of other types of modalities. This multimodal fusion refines the quality of brain images by reducing redundancy and randomness, which contributes to improving the clinical reliability of the diagnosis compared to the use of a single modality. The purpose of this article is first to determine the main steps of the CAD system for brain magnetic resonance imaging (MRI). Then to bring together some research work related to the diagnosis of brain disorders, emphasizing AD. Thus the most used methods in the stages of classification and brain regions segmentation are described, highlighting their advantages and disadvantages. Secondly, on the basis of the raised problem, we propose a solution within the framework of multimodal fusion. In this context, based on quantitative measurement parameters, a performance study of multimodal CAD systems is proposed by comparing their effectiveness with those exploiting a single MRI modality. In this case, advances in information fusion techniques in medical imagery are accentuated, highlighting their advantages and disadvantages. The contribution of multimodal fusion and the interest of hybrid models are finally addressed, as well as the main scientific assertions made, in the field of brain disease diagnosis.

**Keywords:** neuroimaging; Alzheimer's disease; computer-aided diagnosis system; structural and functional imaging; segmentation and classification techniques; multimodal fusion techniques

## 1. Introduction

In the fields of research in medical imaging and diagnostic radiology, computer-aided diagnosis (CAD) has had major interest and development during the last two decades [1–9]. The objective of this

technology is to support radiologists using computer systems in their interpretation of brain images and in the diagnosis of brain diseases. The CAD system provides a second opinion, it makes it possible to analyze medical images thanks to its techniques of pattern recognition and machine learning. This alleviates the fatigue of the radiologist and the burden of the workload, due to the overloaded data. As a result, this technology has the ability to improve diagnostic consistency and accuracy in order to decrease the rate of false negatives, including estimating the extent of the disease.

Alzheimer's disease (AD) is one of the brain disorders that is extremely difficult to identify. It is linked to structural atrophy, pathological amyloid deposits and metabolic alterations in the brain [10]. This neurodegenerative disease is the cause of 60% to 70% of cases of dementia [11], which generally begins slowly and worsens over time. It gradually deteriorates cognitive and behavioral capacities and the causes of this disease remain unknown [1], with the exception of certain hereditary forms. A CAD system can help to perform an early diagnosis which is crucial for mitigating the effects of AD. In fact, several diagnostic tools and approaches have been developed in order to provide measures that make it possible to detect early changes during subclinical periods, clarify the underlying mechanisms and inform neuro-protective interventions aimed at slowing down the extent of the disease. As a result, the rising costs are reduced for families and society.

In addition to this, magnetic resonance imaging (MRI) [12] has been used for a long time to eliminate several sources of brain disorders. This technology provides a detailed description of the anatomy including brain pathology with spatial resolution and soft tissue contrast. It makes it possible to study the structural and chemical correlates of a disease, which improves understanding of the mechanisms involved [13]. In AD, MRI adds a positive predictive value to the diagnosis [5]; solid experiments have shown that changes in brain structure can be detected with structural MRI in elderly subjects with mild cognitive impairment (MCI) [14]. In this context, patients with MCI to be converted to AD are characterized by significant atrophy of the medial temporal lobes, posterior cingula, lateral and parietal temporal cortex compared to control subjects or stationary MCIs.

However, the benefits of structural MRI are limited and several problems have arisen as a result. The MRI went from two-dimensional (2D) to three-dimensional (3D) modality, which caused the neuro-radiologist to rapidly increase the data to be analyzed. Added to this, the resolution and the signal-to-noise ratio (SNR) have become higher [15]. In this regard, the problem of developing new CAD tools, such as data fusion techniques, has been widely addressed in recent years in order to reduce the workload, paying particular attention to the study with a non-invasive way of the existing connectivity between anatomical and functional imaging. This process of data fusion in a multimodal environment makes it possible to generate a more informative merged image which helps diagnosis and forecasting, by combining complementary and redundant information coming from the MRI and from functional modalities, such as, computerized tomography (CT), single photon emission tomography (SPECT) and positron emission tomography (PET), which are characterized by different objectives in radiology. As a result, the resulting fusion image is better suited for visual perception and image processing and analysis tasks [16,17], providing more condensed and relevant information. In addition, instead of storing several multi-source images, a single merged image is taken into account, which reduces memory costs. Additionally, noting that the combination of medical images can often lead to additional clinical information that does not appear in the separate images.

The objective of this review article is to highlight the interest shown in multimodal fusion by researchers in neuroimaging for the diagnosis of brain disorders, in particular AD. It consists of summarizing and examining the main applications, results, perspectives as well as the advantages and disadvantages of different MRI neuroimaging technologies for the diagnosis of brain disorders, with emphasis on the application of multimodal fusion, especially for the diagnosis of AD.

Firstly this review study concerns the collection of several works related to CAD systems for the diagnosis of cerebral dementia, notably AD. We analyze their proposals which were introduced with the hope of helping the radiologist to properly assess the extent of the disease, by providing him with a second opinion in the form of a computer output. In order to reduce the rate of false negatives

and improve the accuracy of the diagnosis, the researchers developed techniques notably in the two main phases in the CAD system, namely segmentation of the brain regions and classification. In this context, we inspect these techniques from artificial intelligence by highlighting their advantages and disadvantages. In addition, we identify the various measurement parameters that have been used to quantitatively assess the performance of the proposed CAD systems. For the classification process, several measurements were considered, such as the sensitivity (SE), which represents the true positive rate; specificity (SP), which estimates the true negative rate; and accuracy (AC), which determines the proportion of true results in the database, whether true positive or true negative. For the same purpose, for some work, the area under the ROC curve (AUC) value was estimated, which determines the diagnostic validity by combining sensitivity and specificity. Likewise, in the segmentation process various measures have been identified, which in the majority take into account the portions of the rates of true and false positives as well as true and false negatives, such as the Tanimoto and dice coefficients, the Jaccard similarity index, etc.

Then, secondly, after identification of the gaps raised by mono-modal MRI CAD systems, we determine one of the solutions proposed in the literature within the framework of the use of data fusion techniques which can be applied in a multimodal imaging environment. The effectiveness of this approach is linked to its power to reconstruct and predict missing information from the MRI. Therefore, the most used fusion techniques are described, highlighting their advantages and disadvantages. Likewise, some key works in the literature are described, which have used multimodal fusion to improve the performance of conventional CAD systems. In addition, a performance evaluation and comparison with single-modal systems is proposed by applying reliability estimation methods such as cross-validation. The measurement parameters are also determined for the purpose of quantitatively evaluating the effectiveness of multimodal CAD systems. Finally, a discussion is tackled which highlights the interest of combining several techniques in the framework of hybrid models and the contribution of multimodal fusion and its usefulness in clinical studies.

The rest of the article is organized as follows: In Section 2, first, we describe the foundations of a CAD system, then we analyze the research work already carried out using the MRI mono modality. In this regard, the methods used in the different phases of the CAD system, including the classification and segmentation of brain regions, are described, summarizing their disadvantages and advantages. We then clarify the problem addressed by the works examined by reporting the solution proposed in the literature. Consequently, the efforts investigated to find a solution within the framework of multimodal fusion of brain images are presented in Section 3. We review the aspects of data fusion, with the aim of providing an overview of the applicability and progress of fusion techniques in medical imaging. In this regard, several research works using multimodal fusion are examined. A performance study of the works selected in the context of experimental classification is presented; and the comparison of the results with those of CAD systems exploiting a single MRI modality is proposed. A discussion is then suggested, summarizing the disadvantages and advantages of the multimodal fusion methods. Concluding remarks as well as ideas and directions for future research are presented in Section 4.

## 2. Cad Systems of Brain Disorders Based on MRI Technology

### 2.1. CAD System Architecture

The architecture of a CAD system associated with a brain image is illustrated in Figure 1, in which several processes are carried out. First, the image from MRI is proposed as an input for the CAD system for which it selects the training samples. Then, a preprocessing and a definition of region(s) of interest (ROI) (block A in Figure 1) are developed to eliminate samples not relevant for the diagnosis. An extraction of the characteristic parameters of the voxels (block B) is carried out thereafter. Finally, segmentation and classification are performed. The segmentation (block C block corresponds to the Section 2.2) groups the voxels into regions, based on the characteristics of the cerebral image. While the classification (block D block corresponds to the Section 2.3) allows to classify the images in two classes:

normal or abnormal. Various machine learning tools have been used successfully in both of these processes, and many artificial intelligence techniques have been developed in the past two decades.

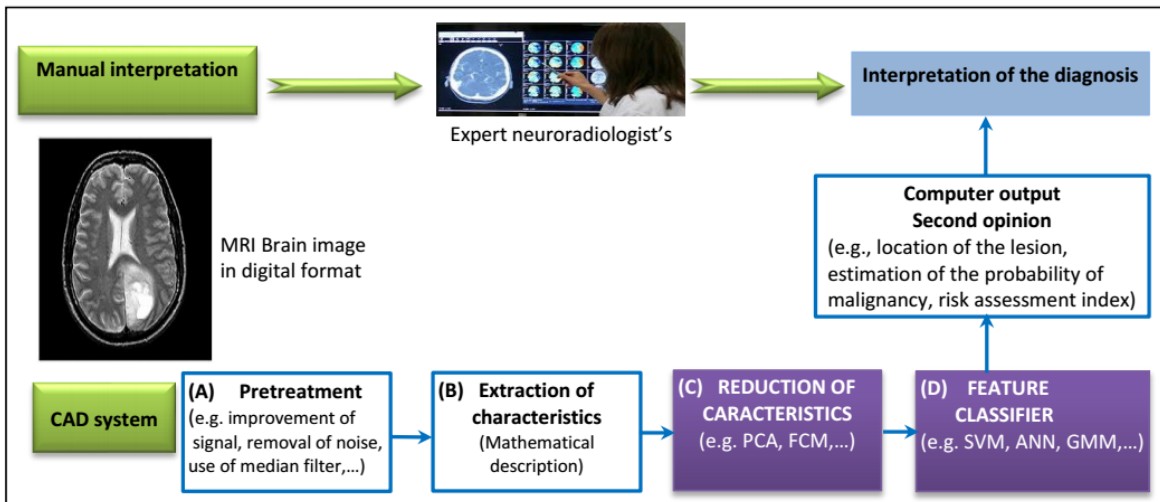

**Figure 1.** Schematic diagram for magnetic resonance imaging (MRI) brain image interpretation (top) and overall block diagram of the corresponding computer-aided diagnosis (CAD) system (bottom). (A) Enhancement of brain signal and definition of regions of interests (ROIs). (B) Extraction of voxel features in a mathematical representation. (C) Reduction of voxel parameters and segmentation in brain regions. (D) Classification and categorization of patients into normal or abnormal classes.

In the following, we are interested in the brain regions segmentation which represents a key step in the CAD system, for which good or bad segmentation determines the success or failure of the next step i.e., the classification process. In this context, there is a large amount of work on neuroimaging and several studies relating to AD, thus various conventional machine learning tools have been used with success and numerous artificial intelligence techniques have been developed.

### 2.2. Segmentation of Brain Regions

Accurate quantification of the volume of brain tissue, particularly cerebrospinal fluid (CSF), gray matter (GM) and white matter (WM) would aid in the diagnosis and understanding of certain neurodegenerative diseases such as AD and Parkinson's syndrome. In this section, which corresponds to block C in Figure 1, we will detail the advances in research in the context of the brain regions segmentation.

#### 2.2.1. Related-Work to the Segmentation of Brain Regions

In [18] the authors proposed the adaptive fuzzy C-means algorithm (AFCM) for the segmentation of multi-spectral MRI images in 2D and 3D. AFCM makes it possible to model the intensity inhomogeneity as a gain field by gradually varying the intensities in the image space. Performance comparisons were made with the fuzzy and noise-tolerant adaptive segmentation method (FANTASM), in a noisy environment. Misclassification rate and mean-squared error are used as evaluation measuring parameters.

In [19], a modified FCM (MFCM) algorithm is proposed which incorporates both the local spatial context and the non-local information by using a new dissimilarity index instead of the usual distance metric. The efficiency of the algorithm is demonstrated by segmentation experiments and by comparison with other advanced algorithms, namely: standard FCM, spatial FCM (FCMS), FCM with spatial information (FCMSI) and fast generalized FCM (FGFCM). The measurement parameters used to quantitatively assess the performance are: Similarity index, false positive ratio and false negative ratio.

In [20], the authors proposed a method which makes it possible to divide the brain into homogeneous regions for the detection of tumors. The process was split into two stages: Pre-segmentation and segmentation and several techniques have been exploited such as anisotropic filtering and the stochastic model Markov random field (MRF). The maximum posterior criterion (MAP) was used to estimate the MRF achievement taking into account the observed dependent data.

In [21] the authors applied the discriminant random fields (DRF) models for the segmentation of brain tumors. A comparison was made with the MRF models using the Jaccard similarity coefficient.

In [22], the authors presented a method for the segmentation of WM tissue lesions (WML). Support vector machines (SVM) were used to integrate characteristics of 4 MRI acquisition protocols to distinguish WML from normal tissue. A visual assessment was performed using two experienced neuro-radiologists. A quantitative validation was also carried out with Pearson correlation, Spearman correlation, coefficient of variation and reliability coefficient.

In [23], the authors proposed the adaptive mean shift (AMS) method to classify voxels in one of the GM, WM and CSF tissues. A comparison was made with the adaptive MAP (AMAP) and maximum posterior marginal-MAP (MPM-MAP) methods. Tanimoto coefficient was applied as an evaluation criterion.

In [24], the authors used sets of contours of multiple sclerosis lesions (MS) taken from MRI segmented images, and united their 3D surfaces by spherical harmonics. The objective was the 3D reconstructions of MS lesions and calculates their volumes. A comparison was made with the slice stacking technique by applying quantitative measurement parameters such as misclassification rate, mean-squared error.

In [25], the authors proposed a multi-context wavelet-based thresholding (MCWT) method to classify pixels with GM, WM and CSF tissues. A comparison was made with the wavelet and multigrid wavelet transforms.

In [26], the authors proposed an algorithm based on the transformation into spherical wavelets. The algorithm is applied to the caudate nucleus and the hippocampus for the study of schizophrenia. The validation performed using the Average max error, average min error evaluation criteria showed efficiency from the calculation point of view and compared to the active shape model algorithm by capturing finer shape details.

In [27], the authors introduced a threshold-based scheme that uses level set (LS) for 3D segmentation of the brain tumor. Two threshold update systems have been developed, based on research and adaptation. The experimental results by applying the following evaluation criteria— Jaccard measure, Hausdorff distance and mean absolute surface distance—demonstrated the effectiveness of the method and its performance compared to the method based on competition by region.

In [28], the authors proposed a method based on neighborhood hypergraph partitioning. The experiments have demonstrated the proper functioning of the method and its performance compared to the normalized cut Ncut) algorithm.

In [29], the authors used the adaptive graph cut method optimized in an iterative mode for the automatic segmentation of MRI brain images. A comparison with conventional graph cut and the MRF model was performed using the classification rate evaluation criterion.

In [30], the authors used the FCM algorithm to combine the average filter with the local median filter in order to perform local segmentation of brain MRI volumes. Tagging is achieved by tagging by region using genetic algorithms (GAs), followed by an amendment in terms of voxel using the growth of parallel regions. The fuzzy model is used both to design the fitness function of GAs and to guide the growing region. Several measurement parameters were applied such as mean, standard deviation, false positive ratio, false negative ratio, similarity index and Kappa statistic.

In [31], the authors proposed a brain tissue segmentation method which aims to calculate the fuzzy membership of each voxel to indicate the degree of partial volume using fuzzy Markov random segmentation. The average error rate was used to assess the performance of the segmentation.

In [32], the authors proposed an adaptive mean-shift algorithm for tissue segmentation in WM, GM and CSF. The Bayesian model was applied to estimate the bandwidth of the adaptive nucleus and to study its impact on the precision of tissue segmentation. A comparison was completed with the hybrid *k*-NN/AMS model using the Dice and Tanimoto coefficients as an evaluation criterion.

In our previous work, hybrid model based brain tissue segmentation was proposed, for images from patients with AD. The approach mainly uses clustering techniques from fuzzy logic in particular, the possibility theory. In [33], the possibilistic C-means algorithm (PCM) was applied to derive fuzzy maps of the volume of WM, GM and CSF tissues, based on an initial partition of tissue centers provided using the FCM.

To make PCM based tissue quantification more robust to noise and artifacts, the FCM algorithm was replaced in [34] by the bias correction FCM (BCFCM) algorithm. Whereas in [35] the FCM partition was optimized, by a genetic process that uses GA.

To ensure the robustness against noise, for the segmentation process based on the hybrid possibilistic-fuzzy-genetic model, intensive experiments were proposed in [36] applied in real and synthetic images, with high additive noise levels which reached 20%.

For the purpose of improving performance and reducing noise sensitivity, experiments were carried out by applying the fuzzy possibilistic C-means (FPCM) algorithm in [37] and the possibilistic fuzzy C-means algorithm (PFCM) in [17] to derive the fuzzy tissue maps. Comparisons were made with FPCM, PCM, FCM and many hybrid clustering algorithms, by applying the Tanimoto coefficient, Jaccard similarity index, specificity and sensitivity.

Table 1 summarizes certain works that have used segmentation techniques from artificial intelligence and applied to brain images.

**Table 1.** Related work to the MRI brain regions segmentation using some techniques described in the literature: The efficiency of the proposed automatic segmentation algorithm is demonstrated by experiments applying quantitative measurement parameters (last column of the table) for the evaluation, and by comparison with visual observation of a clinical expert (manual segmentation) or with other state-of-the-art algorithms (penultimate column).

| Reference | Segmentation | Database | Comparison | Measuring Parameters |
|---|---|---|---|---|
| **Region Approach Based Techniques** | | | | |
| [20] | MRF | Several GD-TI, TI and T2-weighted from radiology department of the Poitiers's hospital | Clinical expert evaluation | Information criterion, MAP criterion |
| [18] | AFCM | Images from Brainweb http://www.bic.mni.mcgill.ca/brainweb | FANTASM | Misclassification rate, mean-squared error |
| [21] | DRF | Several $T_1$, $T_1$c, and $T_2$ images from 7 patients. | MRF | Jaccard similarity index |
| [22] | SVM | Several $T_1$, $T_2$, PD and FLAIR-weighted of 45 diabetics. | Two experienced neuro-radiologists | Pearson correlation, Spearman correlation, coefficient of variation, reliability coefficient |
| [19] | MFCM | Simulated and real MR images. | FCM, FCMS, FCMSI and FGFCM | Similarity index, false positive ratio, false negative ratio |
| [23] | AMS | Images from Brainweb and IBSR http://www.cma.mgh.harvard.edu/ibsr/ | Adaptive MAP, MPM-MAP | Tanimoto coefficient |
| **Form Approach Based Techniques** | | | | |
| [25] | Multi-context wavelet | Several T1-weighted with more than 150 slices for each | Wavelet, multigrid wavelet | NA |
| [26] | Spherical wavelet | Coronal SPGR images with 124 slices for each of 29 left caudate nucleus structures and 25 left hippocampus structures | Active shape model algorithm and expert neuroanatomist evaluation | Average max error, average min error |
| [24] | Spherical harmonics | PD images of multiple sclerosis patients | Slice stacking technique | Mean error and standard deviation. ANOVA criterion |
| [27] | Level sets | Images of 16 patients from Singapore National Cancer Center | Region-competition | Jaccard measure, Hausdorff distance, mean absolute surface distance |
| **Graph Theory Based Techniques** | | | | |
| [28] | Hypergraph | Image with 265 × 256 pixel slices | Ncut algorithm | NA |
| [29] | Adaptive graph cut | Ten images from the Brainweb | Adaptive MRF-MAP, graph cut | Classification rate |

**Table 1.** *Cont.*

| Reference | Segmentation | Database | Comparison | Measuring Parameters |
|-----------|--------------|----------|------------|---------------------|
| **Hybrid Approach Based Segmentation** | | | | |
| [30] | FCM/GA | Several images from Talairach stereotaxic atlas | Manually labeled images | Mean, standard deviation, false positive ration, false negative ration, similarity index and Kappa statistic |
| [31] | Fuzzy/MRF | Images from Brainweb and T1-weighted SPGR | Clinical experts evaluation | Average error rate |
| [32] | Bayesian/AMS | Images IBSR and Brainweb | *k*-NN/AMS | Dice and Tanimoto coefficients |
| [33] | PCM/FCM | PET and T1-weighted from ADNI (http://adni.loni.usc.edu/) | PCM, FCM | Tanimoto coefficient, specificity and sensitivity |
| [34] | PCM/BCFCM | T1-weighted, PET and SPECT scans from Gabriel Mont pied Hospital, France | PCM, FCM and hybrid PCM/FCM | Tanimoto coefficient, specificity and sensitivity |
| [35] | PCM/FCM/GA | PET and T1-weighted from ADNI | PCM, FCM and hybrid PCM/FCM | Tanimoto coefficient, specificity and sensitivity |
| [36] | PCM/FCM/GA 20% of noise | T1-weighted, PET and SPECT scans from Gabriel Mont Pied Hospital, France and ADNI | PCM, FCM | Tanimoto coefficient, specificity and sensitivity |
| [37] | FPCM/FCM/GA 20% of noise | SPECT, PET and T1-weighted from Gabriel Mont Pied Hospital and ADNI | PCM, FCM and hybrid PCM/FCM | Tanimoto coefficient, specificity and sensitivity |
| [17] | PFCM/BCFCM/GA 20% of noise | PET and T1-weighted from Gabriel Mont Pied Hospital and ADNI | FPCM, PCM, FCM and many hybrid clustering algorithms | Tanimoto coefficient, Jaccard similarity index |

MRF: Markov random field, SVM: Support vector machines, FCM: Fuzzy C-means, AFCM: Adaptive FCM, MFCM: Modified FCM, FCMS: FCM spatial, FCMSI: FCM with spatial information, FGFCM: Fast generalized FCM, AMS: Adaptive mean-shift, MAP: Maximum a posteriori probability, MPM: Maximum posterior marginal, FANTASM: Fuzzy and noise-tolerant adaptive segmentation method, DRFs: Discriminative random fields, GA: Genetic algorithm, *k*-NN: *k*-nearest neighbor, PCM: Possibilistic c-means, BCFCM: Bias-corrected FCM, FPCM: Fuzzy PCM, Gd-TI: Gadolinium-titanium, PD: Proton density, FLAIR: Fluid attenuation inversion recovery, T1c: T1 after injecting contrast agent, SPGR: Spoiled gradient recall, NA: Not available.

The techniques most used in the literature are summarized below.

2.2.2. Segmentation Techniques Proposed in Literature: Description, Advantages and Disadvantages

Despite decades of research, there is no standard method that could be considered effective for all types of medical images. However, a set of ad hoc methods have received a certain degree of popularity presented in Figure 2.

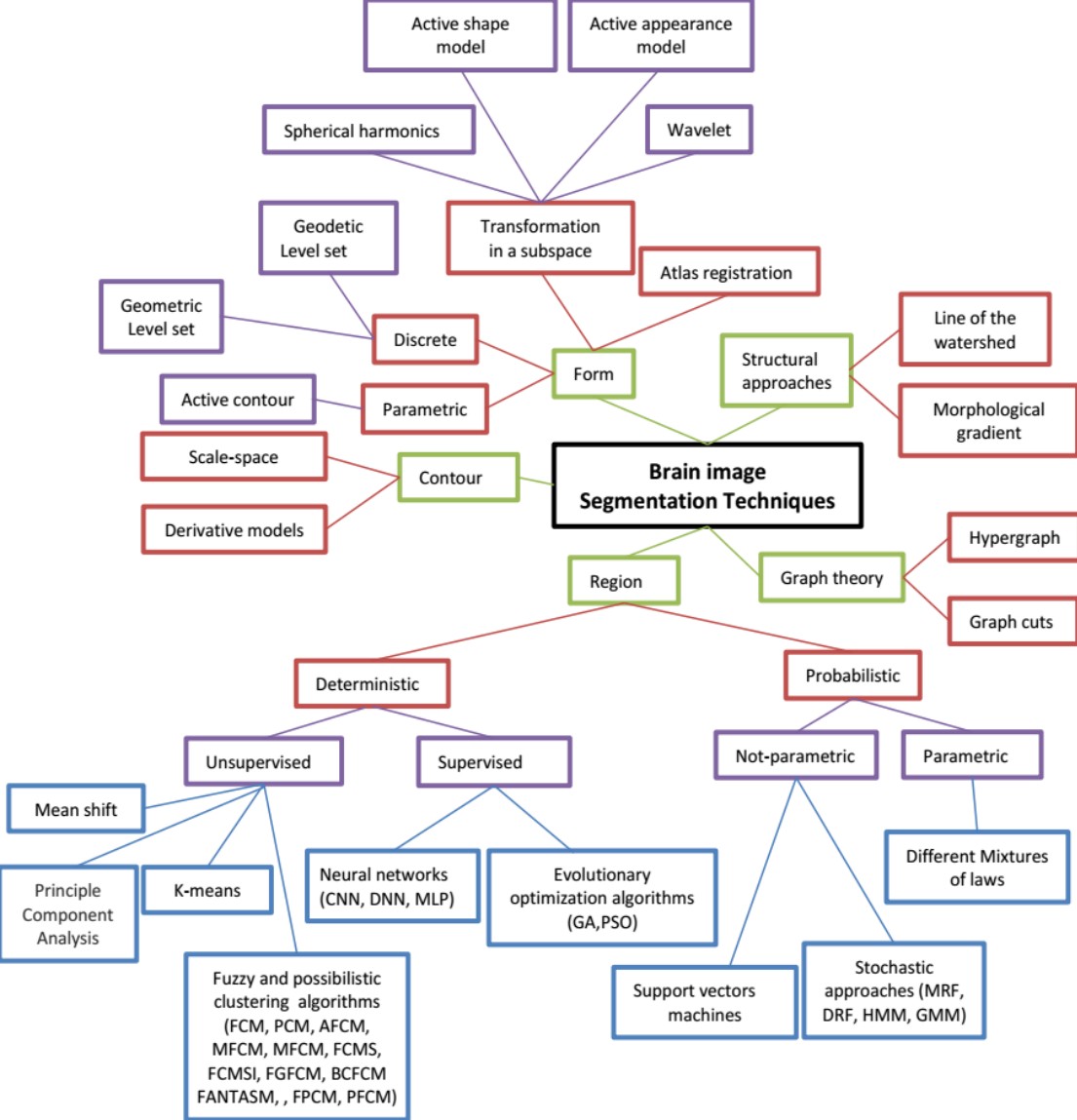

**Figure 2.** Classification of the various MR brain image segmentation methods described in the literature. In this work, the methods were classified in five categories based on the approaches (in green color): Form, structural approaches, graph theory, region and contour. FCM: Fuzzy C-means, AFCM: Adaptive FCM, MFCM: Modified FCM, FCMS: FCM spatial, FCMSI: FCM with spatial information, FGFCM: Fast generalized FCM, FANTASM: Fuzzy and noise tolerant adaptive segmentation method, PCM: Possibilistic C-means, BCFCM: Bias-corrected FCM, FPCM: Fuzzy PCM, GA: Genetic algorithm, PSO: Particle swarm optimization, MRF: Markov random field, DRFs: Discriminative random fields, HMM: Hidden Markov models, GMM: Gaussian mixture model, DNN: Deep neural network, CNN: Convolutional neural network, MLP: Multi-layer perceptron.

The Region Approach

It is designed to partition images into several classes.

- **Artificial neural networks:** ANNs used in several works related to neuroimaging [38–46] are a supervised deterministic method, represented by an interconnected group of artificial neurons using a mathematical model to process information. It performs well in complex and multivariate nonlinear domains, such as tumor segmentation where it becomes difficult to use decision trees or rule-based systems. It also works a little better for noisy data. Data allocation is not required as in the case of statistical modeling. Its learning process consumes enough time, usually with gradient-type methods. The representation of knowledge is not explicit, in the form of rules or other easily interpretable. Initialization may affect the result which may cause overtraining.

- **Genetic algorithms:** GAs exploited in many neuroimaging studies [47–52] are a supervised deterministic method of optimizing research that exploits the concepts of natural selection. It differs from traditional optimization methods in four points: (1) It is a parallel search approach in a population of points, thus having the possibility of avoiding being trapped in a local optimal solution. (2) Its selection rules are probabilistic. (3) She works on the chromosome, a coded version of the potential solutions of the parameters, rather than the parameters themselves. (4) It uses the fitness score, obtained from objective functions, without any other derived or auxiliary information. However, its optimization process depends on the fitness function. It is hard to create good heuristics that really reflect our goal. It is difficult to select the initial parameters (the number of generations, the size of the population, etc.).

- **k-means:** *k*-means [53–57] are a deterministic method based on unsupervised learning which makes it possible to divide a set of data into k clusters. It is widely used for brain segmentation with mainly satisfactory results, to overcome the isolated distribution of pixels inside the image segments. Its execution is simple to implement, fast in real time and in calculation even with a large number of variables. Unfortunately, the unstable quality of the results prevents its application in the case of automatic segmentation. Generally, a degradation of the quality of the segmentation is observed in the case of an automatic segmentation, or when the weight of the pixels in the neighboring local regions is added. Difficulty predicting the k value. Different initial partitions can result in different final clusters. The algorithm only works well when spherical clusters are naturally available in the original data.

- **Fuzzy C-means:** FCM [19,53,58–64] is an unsupervised deterministic method which represents the advanced version of *k*-means. It is based on the theory of fuzzy subsets giving rise to the concept of partial adhesion based on the membership functions. It is widely used in the segmentation and diagnosis of medical images. It provides better results for overlapping data. Unlike *k*-means where data must systematically belong to a single cluster, FCM assigns a fuzzy degree of belonging to each cluster for each data, which allows it to belong to several clusters. However, the computation time is considerable. It does not often provide standard results due to the randomness of the initial membership values. In addition, MRI images often contain a significant amount of noise, resulting in serious inaccuracies in segmentation. It only takes into account the intensity of the image, which causes unsatisfactory results for noisy images. The counterintuitive form of class membership functions limits its use.

- **Mean shift:** it's an unsupervised deterministic method [62,65–67], which makes it possible to locate the maxima—the modes—of a density function, from discrete data sampled through this function. It is based on a non-parametric algorithm which does not take any predefined form on the clusters and assumes no constraint on the number of clusters. Robust with outliers and able to manage arbitrary function spaces. However, it is sensitive to the selection of the window h which is not trivial. An inappropriate window size may result in the merging of modes or the generation of additional "shallow" modes. Costly from a calculation point of view and does not adapt well to the size of the function space.

- **Threshold-based techniques:** the easiest way is to convert a grayscale image to a binary using a threshold value [68]. Pixels lighter than the threshold are white pixels in the resulting image and darker pixels are black pixels. Several improvements have been reported in which the threshold is selected automatically. They are very useful for the linearization of images, an essential task for any type of segmentation. They work well for fairly noisy images. Do not require prior image information. They are useful if the brightness of objects differs significantly from the brightness of the background. Their speed of execution is quite fast with minimum IT complexity. However, do not work properly for all types of MRI brain images, due to the large variation between foreground intensities and background image intensities. The reason why, selecting the appropriate threshold value is a tedious task. In addition, their performance degrades for images without apparent peaks or with a wide and flat valley [69].

- **Region growing method:** it allows to group pixels together in a homogeneous region, including growing, dividing and merging regions. It correctly separates the regions with the same characteristics already defined, especially when the criteria for region homogeneity are easy to define [69–73]. However, it is very sensitive to noise. Costly in memory and sequential from the calculation point of view. In addition, it requires manually selecting an origin point and requires deleting all the pixels connected to the preliminary source by applying a predefined condition.

- **Mixture of laws (Gaussian mixture models):** parametric probabilistic method which allows each observation to be assigned to the most probable class. The classes follow a probability distribution (law), normal in the case of Gaussian mixture models (GMM). GMMs [74–78] require few parameters estimated by a simple likelihood function. These parameters can be estimated by adopting the EM algorithm in order to maximize the likelihood function of the *log*. However, GMM assume that each pixel is independent of its neighbors; this does not take into account the spatial relationships between neighboring pixels [79]. Also, the previous distribution does not depend on the pixel index.

- **Markov Random field:** non parametric probabilistic method, which allows modeling the interactions between a voxel and its neighborhood. In the Markov random field (MRF), the local conditional probabilities are calculated by the Hammersley—Clifford theorem, which allows to pass from a probabilistic representation to the energy representation via the Gibbs field. MRF [80–83] is characterized by their statistical properties; non-directed graphs can succinctly express certain dependencies that Bayesian networks cannot easily describe. It is effectively applied for the segmentation of MRI images, for which there is no natural directionality associated with variable dependencies. In MRF, the computation of the normalization constant Z requires a sum over a number of potentially exponential assignments generally; it is an NP-difficult. In addition, many non-directed models are difficult to interpret or intractable which require approximation techniques.

- **Hidden Markov models:** similar to the MRF, it's non parametric probabilistic method. It is possible to express a posterior probability of a label field from an observation in hidden Markov models (HMM), thanks to the Bayes theorem. HMMs [84–88] make it possible to model arbitrary characteristics of observations, making it possible to inject knowledge specific to the problem encountered into the model, in order to produce an ever finer resolution of spectral, spatial and temporal data. In the case of HMMs, the types of previous distributions that can be placed on masked states are severely limited; it is not possible to predict the probability of seeing an arbitrary observation. In practice, this limitation is often not a problem, as many common uses of HMMs do not require such probabilities.

- **Support vector machines:** SVMs [21,89–92] are a non-parametric probabilistic method whose objective is to find an optimal decision border (hyperplane) which separates the data into groups. The formation process depends on various factors such as the penalty parameter C or the kernel used, such as the linear, polynomial kernel, the radial-based function (RBF) and in its particular case the Gaussian kernel. The generalization performance of this method is high, especially

when the dimension of the function space is very large. It makes it possible to train non-linearly generalizable classifiers in large spaces using a small learning set. It minimizes the number of classification errors for any set of samples. However, it requires a high learning time and memory space for data storage. Moreover, the optimality of the solution can depend on the kernel used unfortunately, there is no theory allowing to determine a priori which will be the best kernel for a concrete task. Also, SVMs assume that the data is distributed independently and identically, which is not appropriate for the segmentation of noisy medical images.

The Contour Approach

The primitives to be extracted are lines of contrast between regions of different gray levels and relatively homogeneous. We could cite the derived models and the scale-space models.

- **Gaussian Scale-Space representation:** this concept [93–96] makes it possible to manage image structures at different scales by generally smoothing. This representation is obtained by solving a linear diffusion equation. Its transparent and natural way of handling scales at the data level makes this concept popular. However, it is sensitive to signal noise since smoothing is applied without an average filter. In addition, parasitic characteristics are to be considered because of the high-frequency noises which introduce local extrema into the signal.
- **Derived models:** they make it possible to model image zones (contours) assuming that the digital image comes from a sample of a scalar function developed with a narrow and differentiable support. In this case, the variations in intensity of the image are characterized by a 3D variable which represents the light intensity corresponding to the illuminations (shadows), changes in orientation or distance, changes in surface reflectance, changes in absorption of rays, etc.

The Structural Approach

It takes into account the structural and contextual information of the image.

- **Morphological gradient:** it's the difference between the operators of the gradient of expansion and erosion of an image [97,98]. In this case, the value of a pixel corresponds to the intensity of the contrast in the nearest neighbor. Generally, the extensive and anti-extensive operators exploited by gradient masks are effective in determining the intensity transitions of gray levels in the borders of objects. However, this technique suffers from the problem of the edge detail smearing. In addition, its sensitivity in particular to white Gaussian noise condensed on the high-frequency part of the signal.
- **Watershed line:** it interprets an image as a height profile, flooded from regional minima so that, the lines where the flooded areas touch represent the watersheds [99,100]. It makes it possible to use the a priori knowledge of the clinician and his intervention, which facilitates visual evaluation in the higher level. However, it is difficult to implement and slow from a calculation point of view. In addition, the over-segmentation of images is generally frequent.

The Form-Based Approach

It searches for areas that derive from a given form a priori.

- **Deformable models:** generate curves or surfaces (from a simple image), or hyper-surfaces (in the case of larger images). They use internal and external forces to delimit the limits of objects and thus distort images. We could distinguish parametric models (active contours or *snake*) [101–105] and geometric models [106]. They are robust to noise and parasitic edges thank to their ability to generate closed parametric surfaces or curves. Simple to implement on the continuum and achieve less than pixel accuracy, a property highly desirable for medical imaging applications. Ease of integrating border elements into a coherent mathematical description. They are able to enlarge

or contract over time, within an image [52,107]. However, they risk producing shapes whose topology is inconsistent with the real object, when applied to noisy images with ill-defined borders.

- **Atlas:** allows an image segmented by an automated algorithm to correspond to a reference image (atlas) [108,109]. These techniques take into account a priori knowledge of brain structures and manage segmentation as a recording problem. They are used in clinical practice, for computer-assisted diagnosis and offer a standard system for detecting properties and morphological differences between patients. They allow segmentation even if there is not a well-defined relationship between the intensities of the pixels and associated regions. However, building an atlas takes time. Difficult to produce objective validation, because segmentation is used when the information from the gray level intensities is not sufficient.

- **Wavelets:** automatically extracts the histogram threshold from the image by wavelet transform. The threshold segmentation is carried out by exploiting multi-scale characteristics of the wavelet transformation [25,56,110–112]. It preserves the sharpness of the contours and provides frequency information located on a function of a signal, which is beneficial for segmentation. However, the overall threshold value is not constant, which leads to a sensitivity of the transformation to the shift. A transformation of dimension greater than 1 suffers from a bad direction when the transformation coefficients reveal only a few orientations of characteristics, in the spatial domain. In addition, there is no information available on the phase of a signal or vector with complex values; it is calculated by applying real and imaginary projections.

- **Spherical harmonics:** It offers solutions of the Laplace equation expressed in a spherical coordinate system [24,113–115]. A base of orthogonal functions is created, which ensures the uniqueness of the decomposition of a form on the unit sphere. So that any finite energy and differentiable function defined on the sphere can be approximated by a linear combination of spherical harmonics. The estimation of the harmonic coefficients makes it possible to model the form with a level of detail relatively linked to the level of the decomposition whose calculation is fast. However, certain continuity constraints should be included when estimating the coefficients. In addition, the results of the shape reconstruction from the decomposition into harmonics are poor when the missing data are concentrated in one area [116].

- **Level set:** consists of representing the segmentation of the contour by the zero level set of a smoothing [108,117–121]. There are two types of methods: geometric and geodesic. Its strong points are the ease of following forms that change topology. Best results for weak and variable signal-to-noise ratios and for non-uniform intensities. Allow to manage any cavity, concavity, convolution, split or fusion. Allow numerical calculations involving curves and surfaces to be performed on a fixed Cartesian grid without having to configure objects. However, the boundaries of the object are not clearly defined by strong image gradients or significant changes in the intensity distribution, this is common in several medical applications, for which image data often suffer from low contrast or fabric noise. In addition, for large images, the execution speed could be very slow and requires manual adjustment of the parameters to obtain optimal results. Possibility of being trapped by an undesirable local minimum which requires additional regularization to obtain the desired minimum.

- **Edge detection techniques:** try to locate points with more or less abrupt changes in gray level [122]. His reasoning is close to human perception and works well for images with good contrast between them [123]. However, its performance degrades in the case of poorly demarcated edges or many edges.

The Theory of Graphs

Mathematical structure allowing to model the relations in pairs between the objects of a set.

- **Hypergraph:** enlargement of the graph, due to its hyper-edges linked with three or more vertices, which is advantageous for processing large data [124,125]. The concept of cross family (intersection

of hyper-edges) derived suits the problem of segmentation on several levels and gives good results. In addition, it provides more meaningful and more robust edge maps. However, its algorithms are quite complex [126].

- **Graph cut:** the image is considered as an undirected graph whose pixels represent the nodes and where the distance between the neighboring pixels forms an edge. A weight is assigned to each edge so that the weight vectors characterize the segmentation parameters [29,127–131]. No initialization is required for this method which guarantees optimal global solutions. Easy to execute and delivers precise results. Ability to integrate constraints and approximate continuous cutting metrics with arbitrary precision. It is applicable for highly textured, noisy, colored images, complex backgrounds, etc. However, it is limited to binary segmentation, and to a special class of functional energy.

In Table 2, some visual segmentation results are reported.

**Table 2.** Visual illustration of the MRI brain segmentation using some techniques described in the literature: The examples illustrate the results obtained with the techniques classified in the five categories, namely: (A) Region approach, (B) form approach, (C) graph theory approach, (D) structural approach and (E) contour approach.

| (A) Region Approach-Based Methods |
|---|

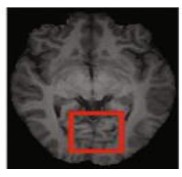 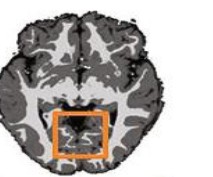 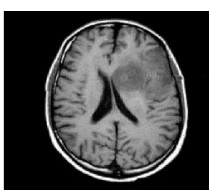 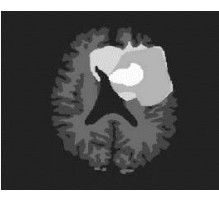 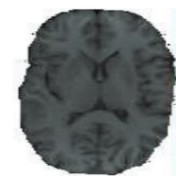 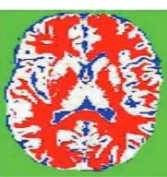

MRI brain segmentation using Convolutional Neural Networks (CNN): (left) Original image, (right) Results of segmentation; illustration reproduced with the permission of Springer Nature [39].

Segmentation of brain tumors from MRI with Support vector machines (SVM): (left) Original MRI, (right) Full brain segmentation; illustration reproduced with the permission of Springer Nature [21].

MRI brain image segmentation with K-Means Algorithm: (left) Original image, (right) Results of segmentation; illustration reproduced with the permission from Elsevier [62].

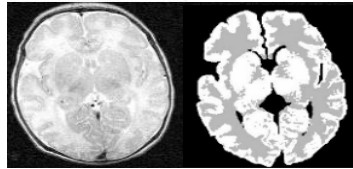 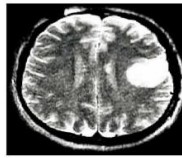 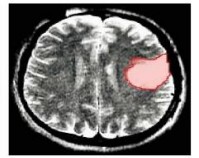 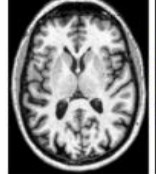 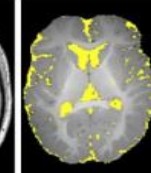 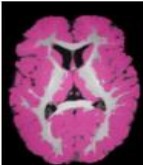 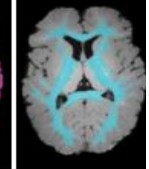

Segmentation of brain MRI image with graph cut approach: (left) Original MRI, (right) Segmentation result; illustration reproduced with the permission of Springer Nature [130].

Segmentation of brain MRI image with Spherical harmonics: (left) frontal oligodendroglioma, (right) the result of segmentation; illustration reproduced with the permission fom Elsevier [113].

Tissue segmentation from MRI using stochastic models: (left to right) Original MRI, CSF, GM, and WM; illustrations reproduced with the permission of PLoS ONE [86].

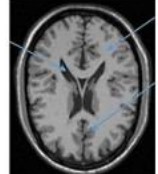 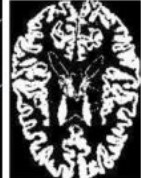 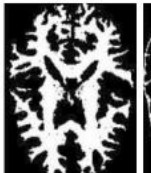 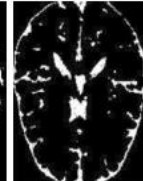 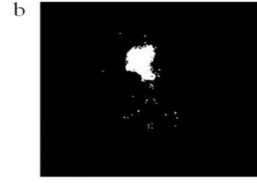 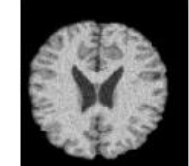 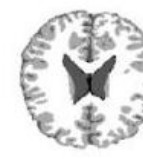 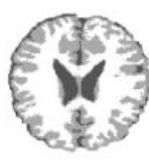

Tissue segmentation from MRI using region growing method: (left to right) Original image, grey matter, white matter and CSF; illustration reproduced with the permission of Springer Nature [132].

Brain tumor segmentation from MRI image using Threshold method: (left) Original image and (right) Segmentation result; illustration reproduced with the permission from Elsevier [133].

Tissue segmentation of brain MRI image with Gaussian Markov Models (GMM): (left) Brainweb sample slice, (middle) Ground truth and (right) Result of segmentation; illustration reproduced with the permission of Springer Nature [134].

**Table 2.** *Cont.*

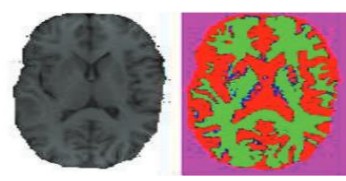
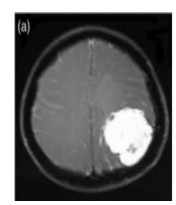
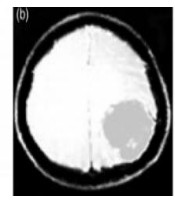
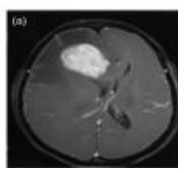
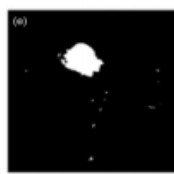

| | | |
|---|---|---|
| Segmentation of brain MRI with means shift: (left) Original MRI and (right) Result of segmentation; illustration reproduced with the permission from Elsevier [62]. | Brain tumor segmentation from MRI image using Fuzzy C-Means (FCM) algorithm: (left) Original image, (right) Results of segmentation; illustration reproduced with the permission of Taylor & Francis Ltd. [50]. | Example of brain tumour segmentation of MRI image using genetic algorithms (GA): (left) Original MRI, (right) the result of segmentation; illustration reproduced with the permission of Taylor & Francis Ltd [50]. |

**(B) Form Approach-Based Methods and (E) Contour Approach-Based Methods**

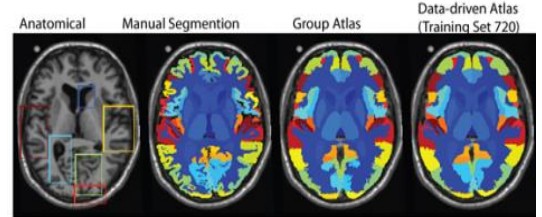
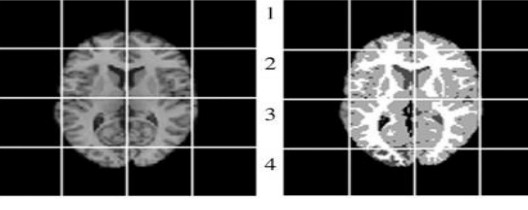
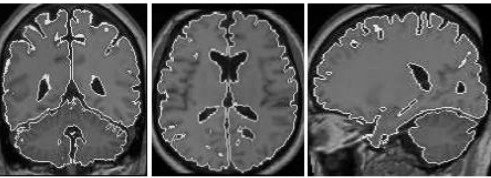

| | | |
|---|---|---|
| (B): Segmentation of brain MRI image with atlas approach; illustration reproduced with the permission of the authors [109]. | (B): Examples of brain MRI image segmentation with wavelet approach: (left) original MRI, (right) the result of segmentation; illustration reproduced with the permission of Elsevier [25]. | (B): Segmentation of brain MRI image with level set of the left hemisphere, right hemisphere and cerebellum; illustration reproduced with the permission of Springer Nature [135]. |


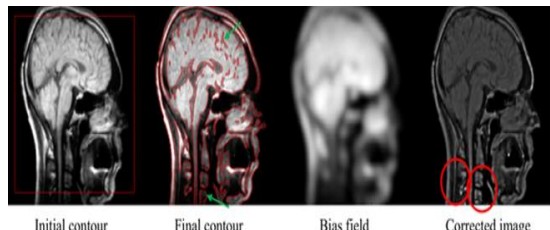
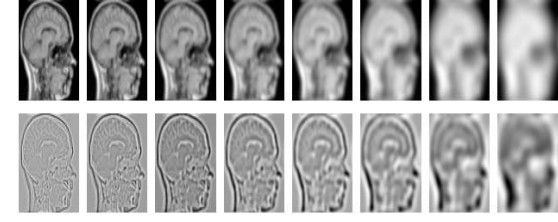

| | | |
|---|---|---|
| (B): Segmentation of white matter in a brain MR image with level set approach: Temporal ordering is from top to bottom, left to right; illustration reproduced with the permission of the author [118]. | (B): Example of brain MR image segmentation and bias correction with active contour approach; illustration reproduced with the permission of PLoS ONE [101]. | (E): Multi-scale representation of an MRI brain image: (top row) Gaussian blur scale-space of a sagittal MRI, resolution 1282, (bottom row) Laplacian scale-space of the same image, same scale range; illustrations reproduced with the permission of Springer Nature [93]. |

**Table 2.** *Cont.*

| (C) Graph theory based methods and (D) Structural approach based methods |
|---|

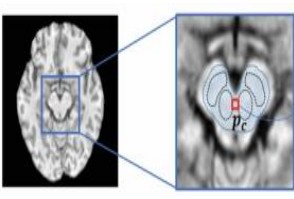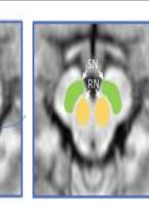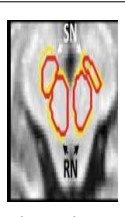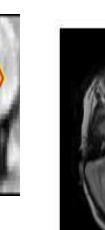

(C): The Brainstem Nuclei segmentation result with hypergraph approach: (left to right) Original image, Brainstem Nuclei, predicted labels and segmentation result (automatic segmentation in red contours and manual segmentations yellow contours); illustration reproduced with the permission of Springer Nature [136].

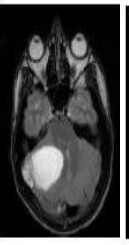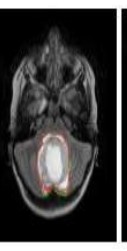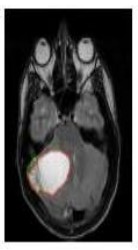

(C): Discriminant model-constrained graph cuts approach to fully automated pediatric brain tumor segmentation in 3-D MRI; illustration reproduced with the permission of Springer Nature [131].

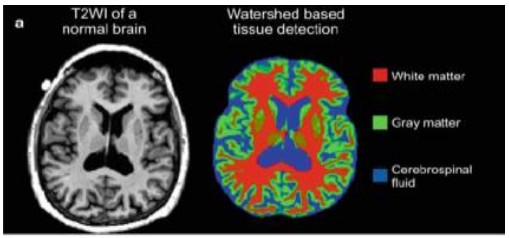

(D): Automated tissue segmentation of brain MRI parsing with line of watershed: (green) gray matter, (red) white matter, (blue) cerebrospinal fluid; illustration reproduced with the permission of Springer Nature [100].

2.2.3. Overview of Software Toolkits for Segmentation of Brain Images

Due to the increasing volumes of medical images, more efficient segmentation software toolkits have been developed and several free, and powerful cross-platform library for brain image informatics, image analysis (processing, registration, segmentation, . . . ) and three-dimensional visualization are available in open access to physicians, researchers and application development. Below is an overview of some of these tools.

FMRIB Software Library

The FSL (www.fmrib.ox.ac.uk\T1\guilsinglrightfsl) [137] offers statistical tools to analyze the brain imaging data of structural MRI, functional MRI (fMRI) and diffusion tensor imaging (DTI). The majority of these tools are functional via GUI. We could cite *FAST* which allows automatic segmentation into different types of tissue and corrects the bias field; *FLIRT* allows linear registration inter- and intra-modal; *MIST* for multimodal image segmentation; *SUSAN* to attenuate non-linear noise and *BET/BET2* to extract brain from non-brain and to model the surfaces of the skull and scalp.

Insight segmentation and registration ToolKit

The ITK (www.itk.org) is an open-source, cross-platform library for processing, segmentation and registration of medical images. It contains algorithms programmed in C++ and wrapped for Python. The implementation is achieved by applying generic programming through the C++ templates. Moreover, CMake generation environment is used to manage the configuration process. It contains scripts for image processing like Gradient image subjected to a Gaussian filter, for the development of segmentation methods such as the region growing method.

ITK-SNAP

The ITK-SNAP (http://www.itksnap.org/) is a software application that offers semi-automatic segmentation of medical images in structures and in three dimensions. It applied active contour approach and allows different tasks namely: seamless three-dimensional image navigation, manual delineation in three orthogonal planes simultaneously of anatomical regions of interest; multiple 3D image formats are considered including NIfTI and DICOM; takes into account the simultaneous linked display, and segmentation of several images; takes into account color, multi-channel images which vary over time and much more...

FreeSurfer

The FreeSurfer (https://surfer.nmr.mgh.harvard.edu/) is an open source software package developed for processing, visualizing and analyzing structural and functional neuroimaging data from cross-sectional or longitudinal studies. Some of the main tasks include: Skullstripping, image registration, subcortical and cortical segmentation, cortical surface reconstruction, cortical thickness estimation, longitudinal processing, fMRI analysis, tractography, freeview visualization GUI and much more... Among these registration tools, we could cite: mri_robust_register, mri_ca_register mri_robust_template, bbregister, mri_cvs_register, mri_em_register, and others.

Analysis of Functional NeuroImages

The AFNI (https://afni.nimh.nih.gov/) is free, open source software for research purposes, developed through support from the National institute of mental health. Its algorithms are programmed with C, Python, R languages and shell scripts. It allows processing, analyzing, and displaying anatomical and functional MRI data. It runs on Unix systems with X11 and Motif displays.

3D Slicer

The 3D Slicer (https://www.slicer.org/) is an open source software platform which applied medical image informatics and allows image processing and three-dimensional visualization. The SlicerDMRI is an extension of 3D Slicer which provides an enhancement diffusion magnetic resonance imaging (dMRI) software. It allows different tasks namely: Load DICOM and nrrd/nhdr dMRI medical image data; load and save tractography in new DICOM format; visualization and registration of multimodal data exploiting the tools of 3D Slicer; and much more . . .

### 2.2.4. Critical Discussion about Segmentation Techniques

Segmentation is a key step that determines the success or failure of the classification process in the CAD system. However, it represents an arduous process, due to the complexity of the medical data, the diversity of the artefacts in particular, the low signal/noise ratio, the uncertain limits of the images, the great variability of the tissues within the same population and the artifacts due to patient movement or little time for data acquisition.

Despite decades of intensive research in the area of brain region segmentation, there is no reliable general theory for all types of images. That said, no standard method is established or considered effective. Therefore, a set of ad hoc methods has been devised by researchers that have received a certain degree of popularity. We can split these segmentation techniques into five categories: those based on shape, contour, region, graph theory and on the structural approach.

Certain conclusions could be drawn concerning these methods. The neural model suffers from complexity from the point of view of the desired topology, it's appropriate learning and the generalization of the network. It would only be chosen if no prior distribution is required and no very high-quality object information is required. The MRF has been widely exploited due to its use of spectral, spatial, textural, contextual and earlier image properties. However, its implementation is very complex and it does not allow integration of form. The watershed model could however be applied to the segmentation of medical images, however, intensive research is needed to understand its mechanism. By making several models evolve simultaneously, the use of GA makes it possible to remedy the problem of local minima observed in deformable models, the estimation of the pose as well as the initialization of the model. The strong point of the clustering methods based on the theory of fuzzy subsets lies in the resolution of the ambiguities of the borders of the regions. In the framework of hybrid models they combine with neural networks, the MRF model and histogram thresholding method.

The segmentation performance also depends on the homogeneity measures which must be taken into account in the analysis of complex regions. Several have been used in literature, spectral, spatial, texture, shape, scale, size, compactness and contextual, temporal and prior knowledge. Spectral measurement was the most primitive. However it is unable to process high-resolution imagery. Texture measurement was also widely used since it simultaneously benefits from spectral and spatial properties. However it often does not provide perfect segmentation. Therefore, to properly estimate the threshold of homogeneity or heterogeneity of the brain regions, it is beneficial to combine all or most of the measures for better segmentation results [138–141]. For example, the interest of integrating prior knowledge and contextual information has opened up good tracks of research in the segmentation of medical images [47,142].

In addition to this, the required information scale which is important for better segmentation performance unfortunately, it is selected manually in most existing works. In addition, it would be relevant to propose a quantitative analysis to the methods for assessing segmentation.

Moreover, we noticed that some studies have proven the interest of applying hybrid models which integrate the advantages of several intelligent methods derived from soft computing in order to solve certain problems encountered in the brain regions segmentation, in particular the combination of fuzzy logic clustering algorithms, neural networks, stochastic models and bio-inspired optimization

algorithms. The interest of this hybridization is revealed in the context where medical images are susceptible to different artifacts and noises which cause disconnected and indistinct limits.

In the following, an effort to bring together most of the advantages and disadvantages of the techniques proposed for the classification of brain images (bloc D in Figure 1) is summarized below. Thus the reported existing works in the literature, on the diagnosis of brain diseases including AD are summarized.

### 2.3. Classification of Brain Images

Several CAD systems have been proposed to distinguish patients affected by cerebral dementias in particular, those used to predict AD and to distinguish it reliably from normal aging. In this section which corresponds to block D in Figure 1, we will detail the advances in research in the context of the brain images classification. Table 3 explores certain AD related works [7,143–156] described in the literature which have applied classification approaches from artificial intelligence and pattern recognition.

### 2.3.1. Classification Techniques Proposed in Literature: Description, Advantages and Disadvantages

Below is an effort to bring together most of the classification techniques (see Figure 3) used for the diagnosis of AD, emphasizing their advantages and disadvantages.

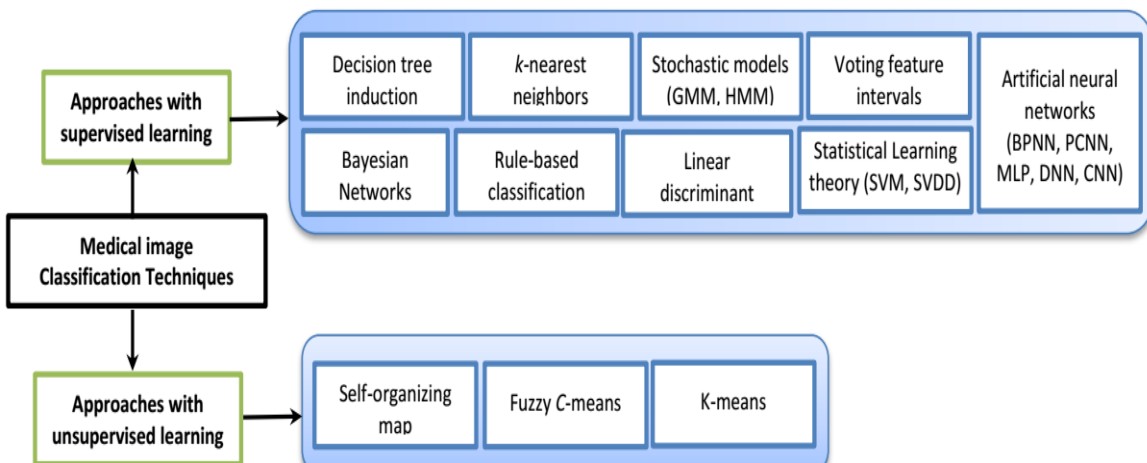

**Figure 3.** MRI brain images classification methods described in the literature: In this work, the classification methods have been grouped into two categories (green color) according to the type of learning, namely: supervised and unsupervised learning. DNN: Deep neural network, CNN: Convolutional neural network, MLP: multi-layer perceptron, BPNN: Back propagation neural network, PCNN: Pulse-coupled neural network, SVM: Support vector machines, SVDD: Support vector data description, GMM: Gaussian mixture model, HMM: Hidden Markov models.

**Table 3.** Some works described in the literature related to the computer aided-diagnosis systems of Alzheimer's disease through MRI. These CAD systems used classification methods based on supervised or unsupervised training. The efficiency of the proposed CAD system is demonstrated by estimating the percentage of the following performance measures (last column): Sensitivity (SE) which represents the true positive rate, specificity (SP) which estimates the true negative rate and accuracy (AC) that determines the proportion of true results in the database, whether true positive or true negative. In [143], the reported learning time was around a week, while in [148] the computation time reported was 0.0451 s per image.

| Reference | Classification Techniques | Database | Performance Measures (%) | | |
|---|---|---|---|---|---|
| | | | AC | SE | SP |
| [143] | Classification: SVM (linear basis kernel) with supervised learning | Group 1: 20 AD and 20 HC, samples from the Rochester community, Minnesota, USA. Group 2: 14AD and 14 HC from the Dementia Research Centre, Univ. College London, UK. Group 3: 33 probable mild AD and 57 HC sample in Rochester, Minnesota, USA. Group 4: 19 subjects with pathologically confirmed FTLD | AD: 96 PMAD: 89 FTLD: 89 | NA | NA |
| [144] | Segmentation: hierarchical networks Classification: SVM with supervised learning | ADNI: 100 P-MCI and 125 HC subjects http://adni.loni.usc.edu/ | MCI: 84.35 | NA | NA |
| [145] | Classification: SVM with leave-1-out CV and 3-fold CV and supervised learning | 19 AD and 20 HC subjects | AD: 80 | NA | NA |
| [146] | Segmentation: SPM5 software from department of Imaging Neuroscience, London, UK and using Student *t* tests | 25 AD subjects (11 men, 14 women), 24 MCI subjects (10 men, 14 women) and 25 HC (13 men, 12 women) subjects | AD: 84 MCI: 73 | 84 75 | 84 70 |
| [147] | Feature extraction: Wavelet coefficients Classification: SOM neural network with unsupervised learning SVM (linear, Polynomial, RBF basis kernel) with supervised learning | AANLIB of Harvard Medical School: 46 AD and 6 HC subjects. http://med.harvard.edu/AANLIB/ | AD-SOM: 94 AD-SVM: 98 | NA | NA |
| [148] | Feature extraction: Wavelet transform Segmentation: PCA Classification: BPNN with supervised learning | 48 AD, glioma, meningioma, visual agnosia, Pick's disease, sarcoma, and Huntington's disease and 18 HC subjects | 100 | NA | NA |
| [149] | Classification: SVM (bootstrap method) with supervised learning | 16 AD and 22 HC subjects | AD: 94.5 | 91.5 | 96.6 |

**Table 3.** *Cont.*

| Reference | Classification Techniques | Database | Performance Measures (%) | | |
|---|---|---|---|---|---|
| | | | AC | SE | SP |
| [150] | Feature extraction: Random forest Classification: SVM (bootstrap estimation and 20-fold CV) with supervised learning | 144 AD and 189 HC subjects | AD: 0.97 | 89 | 94 |
| [151] | Classification: SVM (leave-one-out CV) with supervised learning | 37 AD and 40 HC subjects. | AD:96.1 | NA | NA |
| [152] | Classification: SVM, Bayes statistics, VFI with supervised learning | 32 AD, 24 MCI and 18 HC subjects | AD: 92 MCI-c: 75 | NA | NA |
| [153] | Feature selection: Pearson's correlation Classification: SVM (linear basis kernel and leave-one-out CV) with supervised learning | 20 AD and 25 HC subjects from Hospital de Santiago Apostol, Mexico. | AD: 100 | NA | NA |
| [154] | Features extraction: MBL Classification: LDA, SVM with supervised learning | ADNI: 198 AD, 238 S-MCI, 167 progresses MCI and 231 HC subjects | AD-SVM: 86 AD-LDA: NA | 94 93 | 78 85 |
| [155] | Feature extraction: SIFT Segmentation: k-means Classification: SVM (leave-one-out CV) with supervised learning | 100 AD and 98 HC subjects. | AD: 86 | NA | NA |
| [156] | Feature extraction: fractal analysis Classification: SVM (quadratic kernel) with supervised learning | 13 AD and 10 HC subjects. | AD: 100 | NA | NA |
| [7] | Segmentation: Hybrid FCM/PCM Classification: SVM (RBF kernel and leave-one-out CV) with supervised learning | 45 AD and 50 HC subjects from ADNI phantom with noisiest images and spatial intensity inhomogeneity | AD-MRI: 75 AD-PET: 73 | 84.87 86.36 | 81.58 82.67 |

PMAD: p-mild AD, SVM: Support vector machines, FTLD: frontotemporal lobar degeneration, HC: Healthy control, AD: Alzheimer's disease, ADNI: Alzheimer disease neuroimaging, P-MCI: Probable mild cognitive impairment, S-MCI: Stable MCI, CV: Cross-validation, SPM: Statistical Parametric Mapping, SOM: Self-organizing maps, RBF: Radial basis function, VFI: voting feature intervals, PCA: Principal component analysis, BPNN: Back propagation neural network, AUC: Area under curve, MBL: Manifold-based learning, LDA: Linear discriminant analysis, SIFT: Scale-invariant feature transforms, FCM: Fuzzy C-means, PCM: Possibilistic C-means algorithm.

Techniques based on Supervised Learning

- **Artificial neural networks:** ANNs are used in many works related to neuroimaging as classification method [15,157–163] have been widely applied as a classifier to distinguish new test data. They are universal functional approximations allowing to approximate any function with arbitrary precision. They are flexible nonlinear models for modeling complex real-world applications. They are self-adapting adaptable to data, without explicit specification of the functional or distributional form with the underlying model. They are able to estimate the later probabilities, necessary to establish classification rules and statistical analyzes. However, the learning time is high for large ANNs and the adjustment of the parameters to be minimized requires a lot of calculation.

- **k-nearest neighbors:** the k-NNs proposed in many neuroimaging studies [164–167] allow a test sample to be classified in the class most frequently represented among the k closest training samples. In the case of two or more classes, it will be classified to the class with a minimum average distance. This classifier is powerful and simple to implement. It provides precise distance and weighted average information about the pixels. However, its efficiency degrades for large-scale and large-scale data due to its "lazy" learning algorithm. The choice of k affects classification performance which is slow and the memory cost is high.

- **Gaussian mixing model:** GMM suggested by many neuroimaging researchers [76,168–172] is easy to implement. Effective and robust due to its probabilistic basis. It does not require a lot of time, for large data sets. However, this classifier does not exclude exponential functions, and its ability to follow trends over time is slow.

- **Support vector machines:** SVMs used in several works [6,159,173–176] have high generalization performance, especially when the dimension of the function space is very large. These machines offer the possibility of training generalizable nonlinear classifiers in large spaces using a small learning set. They minimize the number of classification errors for any set of samples. However, learning SVM is slow and requires computation time for implementation. High cost of memory space to store data. No method is approved to determine a priori the best kernel for a concrete task. So the optimality of the solution can depend on the chosen kernel.

Techniques based on Unsupervised Learning

- **Self-organizing map:** SOM [112,176,177] is a type of ANN that produces a discrete, low-dimensional representation of the input space for learning samples. This classifier is simple to implement and easy to understand. Capable of handling various classification issues while providing a useful, interactive and intelligible summary of the data. However, despite the ease of viewing the distribution of input vectors on the map, it is difficult to properly assess the distances and similarities between them. In addition, if the output dimension and the learning algorithms are selected incorrectly, similar input vectors may not always be close to each other and the formed network may converge to local optima.

- **Fuzzy C-means:** FCMs [177–181] make it possible to determine a degree of data belonging to each class. However, it is necessary to set a priori certain parameters, such as the initial partition, the number of classes and their centroids.

### 2.3.2. Critical Discussion about Classification Techniques

Existing work reported in the literature has shown that the classification of brain images is possible via supervised techniques such as ANN, Bayesian networks, k-NN, GMM, HMM, decision tree induction, rule-based classification, PCA and SVM [88,166,182], and via unsupervised techniques such as SOM and FCM. In reality, unsupervised classification, which does not require training data, has not been widely used in CAD systems, due to the specificity of the brain images in which the CAD system should be trained according to the truth field or clinical evidence. On the other hand, the supervised classification was the best adopted because, before the clinical test, the CAD system

introduced characteristic values for the data chosen for learning by doing, teaching the classifier to know the labels of the target class by assigning values binaries (one for the target class and zero for the second class). In this case, in order to improve robustness, the system should be trained with a sufficient amount of training data in order to remedy the problem of overtraining.

Several researchers have been interested in the application of hybrid models which aim to combine the advantages of different intelligent techniques of "soft computing" within the same system, or to combine the relative strengths of different classifiers and apply them in a sequence of so that the overall accuracy is maximized, which allows greater flexibility in modeling dynamic phenomena. However, the cost of computing these systems is sophisticated and high.

Unfortunately, the single-modality MRI-based CAD systems and using these classification techniques have limited performance and cannot provide comprehensive and accurate information especially in real applications where noise and artifacts are condensed. In neuroimaging, the other modalities could provide additional information, whose use is generally necessary for a relevant diagnosis of cerebral dementia. This information is combined using fusion techniques from artificial intelligence that provide for human visual perception a fusion image providing additional and useful clinical information that does not appear in the separate images.

Thus the efforts investigated to find a solution within the framework of multimodal fusion are presented in the following section. We summarize some works related to multimodal fusion, with an experimental performance study and a comparison with systems using a single MRI modality for diagnosis. We also provide an overview of the applicability and progress of information fusion techniques in medical imaging, highlighting the disadvantages and advantages of the methods suggested by researchers in the context of multimodal fusion.

## 3. CAD Systems of Brain Disorders Based on Multimodal Fusion

### 3.1. Motivation for the Application of Multimodal Fusion

Given its clinical accessibility, magnetic resonance imaging technology has been widely used as a non-invasive tool for diagnosing brain diseases because it does not use ionizing radiation, which makes it safe. However, MRI is sensitive to movement, which limits its effectiveness especially in the diagnosis of mobile organs. To overcome this problem and in order to obtain better performance from the CAD system, several researchers have attempted to combine MRI with other modalities using multimodal fusion. With this technology, one could predict and reconstruct missing information that is not available in the MRI. We could also extract additional characteristics, not visible in the MRI images.

Citing, for example, the multimodal CT/MRI fusion, whereby, thanks to the CT image, dense structures (bones and implants) are visualized with less distortion; however, physiological changes are not localized. While the MRI image detects normal and pathological information on the soft tissue, while information relating to the bones is not considered [183]. In addition, the MRI-T1/MRI-T2 fusion, whereby, thanks to the T1-weighted MRI, details on the anatomical structures are provided, while the T2-weighted MRI detects a greater contrast between normal and abnormal tissues. Added to all of this is the MRI/PET merger, whose functional information is extracted using the PET image. This information locates the metabolic changes caused by the growth of abnormal cells before an anatomical abnormality. While the MRI image, thanks to its high-resolution, provides anatomical information about the regions (or tissues) affected by the disease.

The MRI/PET fusion was widely used for the diagnosis of AD. Recent studies [184–188] have shown that this fusion effectively contributes to accurately interpreting the location and extent of AD with combined information. In fact, the MRI measures the early structural changes in the medial temporal lobe, in particular the entorhinal cortex and the hippocampus. Then, PET—FDG (FluoroDeoxyGlucose) [189] makes it possible to observe, in AD patients, the reduction of glucose metabolism in the parietal, posterior cingulate and temporal regions of the brain [190].

Table 4 reports some work related to brain disease diagnostic systems [183,191–202], while Table 5 summarizes some CAD systems related to AD [10,16,17,184–188,203–211], with a comparative study with systems using only MRI for the purpose of exploring the efficiency of multimodal fusion. In this context, the researchers proposed techniques for merging data from artificial intelligence, and applied in a multimodal imaging environment, in order to create an improved fusion image more suited to image processing tasks such as than segmentation and diagnosis. The most widely used fusion techniques in the literature are summarized below.

### 3.2. Multimodal Fusion Techniques Proposed in Literature: Description, Advantages and Disadvantages

Data fusion techniques (see Figure 4) [212–215] can be classified into three categories, depending on the level of the desired fusion: pixel or imaging sensor level, level of functional parameters and level of decision.

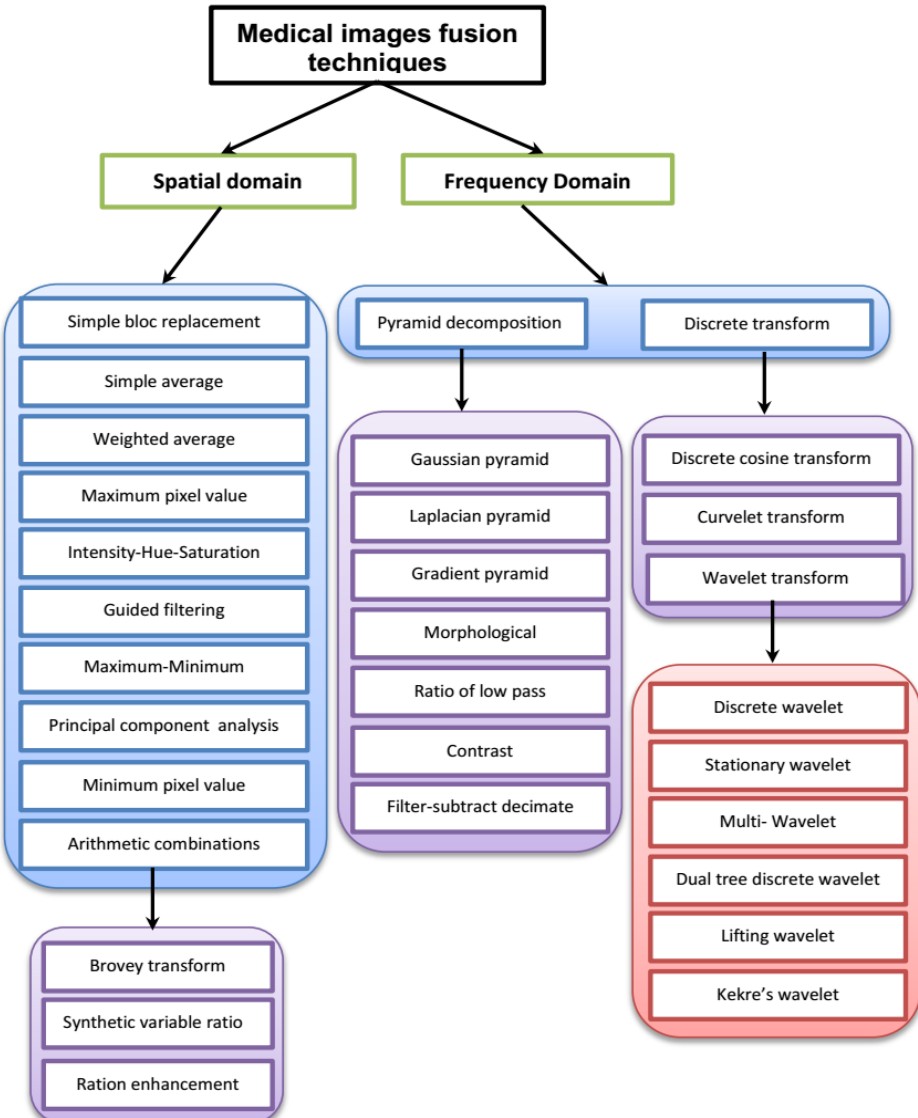

**Figure 4.** Multimodal fusion techniques in spatial and frequency domains for brain images. In this work, image fusion methods have been classified into two categories (green color) according to the domain from which the image is transferred, namely spatial and frequency domains.

**Table 4.** Some works described in the literature related to the computer aided-diagnosis systems of brain diseases through multimodal fusion: The efficiency of the proposed CAD system is demonstrated by experiments applying quantitative measurement parameters (penultimate column of the table) for the evaluation, and by estimating computation time in seconds for some reported works (last column).

| Reference | Fusion Approach | Modalities | Performance Criteria | | | Computation Time (in Sec.) |
|---|---|---|---|---|---|---|
| [191] | Hybrid NSCT/PCNN | | $Q_{MI}$ | $Q_S$ | $Q^{AB/F}$ | |
| | | MRI-$T_1$ + MRI-$T_2$ | 3.9161 | 0.6561 | 0.6841 | 2.2198 |
| | | MRI + CT | 1.8028 | 0.4651 | 0.6652 | 2.2220 |
| [192] | | | $Q_{MI}$ | $Q_S$ | $Q^{AB/F}$ | |
| | NSCT | MRI-$T_1$ + MRI-$T_2$ | 3.9133 | 0.6892 | 0.6961 | 2.2189 |
| | NSCT | MRI + CT | 1.8499 | 0.4703 | 0.6814 | 2.2245 |
| [183] | | | $Q_{MI}$ | $Q_S$ | $Q^{AB/F}$ | |
| | NSCT | MRI-$T_1$ + MRI-$T_2$ | 3.9493 | 0.6950 | 0.6990 | 2.2194 |
| | NSCT | MRI + CT | 1.8503 | 0.4725 | 0.6772 | 2.2198 |
| | PCA | MRI-$T_1$ + MRI-$T_2$ | 3.6627 | 0.6760 | 0.6645 | 0.0333 |
| | PCA | MRI + CT | 2.6001 | 0.5133 | 0.6092 | 0.0328 |
| [193] | | | $Q_{MI}$ | $Q_S$ | $Q^{AB/F}$ | |
| | Contourlet | MRI-$T_1$ + MRI-$T_2$ | 3.8314 | 0.6674 | 0.6816 | 1.9522 |
| | Contourlet | MRI + CT | 1.6025 | 0.4277 | 0.6485 | 1.9682 |
| [194] | | | $Q_{MI}$ | $Q_S$ | $Q^{AB/F}$ | |
| | Wavelet | MRI-$T_1$ + MRI-$T_2$ | 3.0773 | 0.6585 | 0.6176 | 0.0759 |
| | Wavelet | MRI + CT | 1.5420 | 0.4187 | 0.5175 | 0.0780 |
| [195] | | | RMSE | | | |
| | Hybrid Surface/Voxel | MRI + PET | 0.047796 | | | 24.2 |
| | Surface | MRI + PET | 1.69186 | | | 3.4 |
| | Voxel | MRI + PET | 0.050478 | | | 639.18 |

**Table 4.** *Cont.*

| Reference | Fusion Approach | Modalities | Performance Criteria | | | | Computation Time (in Sec.) |
|---|---|---|---|---|---|---|---|
| | | | **PSNR** | | | | |
| [196] | Wavelet | MRI + CT | 72.1172 | | | | NA |
| | Pyramid | MRI + CT | 70.1061 | | | | NA |
| | Weighted average | MRI + CT | 68.4456 | | | | NA |
| | | | **Mean** | **Var** | **Entropy** | **Cros-Ent** | |
| [197] | Weighted average/WT | MRI + CT | 59.6862 | 59.6871 | 6.7599 | 0.5632 | NA |
| | Wavelet | MRI + CT | 32.0674 | 32.0678 | 5.8570 | 0.8999 | NA |
| | Weighted average/WT | MRI + PET | 45.3537 | 45.3543 | 5.6779 | 0.7714 | NA |
| | Wavelet | MRI + PET | 30.1334 | 30.1339 | 5.3272 | 0.9629 | NA |
| | | | **MI** | | | | |
| [198] | Level set | MRI-$T_1$ + MRI-$T_2$ | 1.7176 | | | | NA |
| | Pixel-based | MRI-$T_1$ + MRI-$T_2$ | 1.5540 | | | | NA |
| | Edge detection | MRI-$T_1$ + MRI-$T_2$ | 1.6726 | | | | NA |
| | | | **DC** | | | | |
| [199] | Bayesian multi-sequence Markov model with adaptive weighted EM | MRI-$T_1$ + MRI-$T_2$ + MRI-Flair | 12 (+8) (9% SNR) | | | | NA |
| [200] | | | **EFA** | **EFLA** | **EFSA** | | |
| | SP-MMI algorithm | MRI + SPECT | 0.001 | 0.003 | 0.007 | | NA |
| [201] | | | **EFA** | **EFLA** | **EFSA** | | |
| | FCM/MMI algorithm | MRI + SPECT | 0.06 | 0.16 | 0.04 | | NA |
| | | | **DR** | | | | |
| [202] | Regional growing approach | MRI-$T_2$ + DTI | Edema: 2.96 | | | | NA |
| | | | Tumor solid: 8.07 | | | | |
| | | | Tumor: 11.03 | | | | |

$Q_{MI}$: Information theory-based metrics, $Q_S$: Image structural similarity-based metrics, $Q^{AB/F}$: Image feature-based metrics, NSCT: Non-subsampled contourlet transform, PCNN: Pulse-coupled neural network, PCA: Principal component analysis, RMSE: Root-mean-square-error, PSNR: Peak-to-peak signal-to-noise ratio, sd: Standard deviation, SP-MMI: Surface-projection maximization mutual information, EFA: Error function of area, EFLA: Error function of long-axis, EFSA: Error function of short-axis, FCM: Fuzzy C-means, MI: Mutual information, WT: Wavelet, DC: Dice similarity coefficients, AC: Correct tumor diagnoses accuracy, DR: WM fiber pixel distribution ratio.

**Table 5.** Some works described in the literature related to the CAD systems of Alzheimer disease through multimodal fusion with comparison of performance of single-modal and multimodal classification methods using 10-fold cross-validation. The efficiency of the proposed multimodal CAD system is demonstrated by estimating the percentage of the following performance measures: Sensitivity (SE), which represents the true positive rate; specificity (SP), which estimates the true negative rate; and accuracy (AC), which determines the proportion of true results in the database, whether true positive or true negative. For the same purpose, for some work, the area under ROC curve (AUC) value was estimated which determines the diagnostic validity by combining sensitivity and specificity.

| Reference | Multimodal Classifier | ADNI Subjects | Modalities | Performance Criteria (%) | | | | | | | |
| --- | --- | --- | --- | --- | --- | --- | --- | --- | --- | --- | --- |
| | | | | AD Vs. HC | | | | MCI Vs. HC | | | |
| | | | | AC | SE | SP | AUC | AC | SE | SP | AUC |
| [16] | HIS | NA | MRI + PET | 80-90 | NA | NA | NA | NA | NA | NA | NA |
| [10] | MKL | 51AD + 99MCI + 52HC | MRI | 86.2 | 86.0 | 86.3 | NA | 72.0 | 78.5 | 59.6 | NA |
| | | | CSF | 82.1 | 81.9 | 82.3 | NA | 71.4 | 78.0 | 58.8 | NA |
| | | | PET | 86.5 | 86.3 | 86.6 | NA | 74.5 | 81.8 | 66.0 | NA |
| | | | MRI + FDG−PET | 90.6 | 91.4 | 91.6 | NA | 76.4 | 80.4 | 63.3 | NA |
| | | | MRI + FDG−PET + CSF | 93.2 | NA | NA | NA | NA | NA | NA | NA |
| [203] | MKL | 77AD + 82HC | MRI | 75.27 | 63.06 | 81.86 | 82.48 | NA | NA | NA | NA |
| | | | FDG−PET | 79.36 | 78.61 | 78.94 | 83.9 | NA | NA | NA | NA |
| | | | MRI + FDG−PET | 81.0 | 78.52 | 81.76 | 88.5 | NA | NA | NA | NA |
| [204] | MKL | 48AD + 66HC 48AD + 66HC | MRI + FDG−PET | 87.6 | NA | NA | NA | NA | NA | NA | NA |
| | | | MRI + FDG−PET + CSF + ApoE + Cognitive scores | 92.4 | NA | NA | NA | NA | NA | NA | NA |
| [205] | SCLDA model | 49AD + 67HC | MRI + FDG−PET | 94.3 | NA | NA | NA | NA | NA | NA | NA |
| [206] | M3T | 45AD + 91MCI + 50HC | MRI | 84.8 | NA | NA | NA | 73.9 | NA | NA | NA |
| | | | FDG−PET | 84.5 | NA | NA | NA | 79.7 | NA | NA | NA |
| | | | CSF | 80.5 | NA | NA | NA | 53.6 | NA | NA | NA |
| | | | MRI + FDG−PET + CSF | 93.3 | NA | NA | NA | 83.2 | NA | NA | NA |
| [207] | Multivariate analysis of OPLS | 96AD + 162MCI + 111HC | MRI | 87.0 | 83.3 | 90.1 | 0.930 | 71.8 | 66.7 | 79.3 | 78.26 |
| | | | CSF | 81.6 | 84.4 | 79.3 | 0.861 | 70.3 | 66.7 | 75.7 | 77.06 |
| | | | MRI + CSF | 91.8 | 88.5 | 94.6 | 0.958 | 77.6 | 72.8 | 84.7 | 87.6 |
| [185] | Random forest | 37AD + 75MCI + 35HC | MRI | 82.5 | 88.6 | 75.6 | NA | 67.3 | 64.3 | 73.9 | NA |
| | | | FDG−PET | 86.4 | 85.8 | 87.1 | NA | 53.5 | 42.3 | 78.0 | NA |
| | | | CSF | 76.1 | 72.8 | 79.8 | NA | 61.7 | 61.6 | 61.8 | NA |
| | | | Genetic | 72.6 | 71.3 | 74.1 | NA | 73.8 | 94.7 | 26.6 | NA |
| | | | MRI + PET + CSF + Genet | 89.0 | 87.9 | 90.0 | NA | 74.6 | 77.5 | 67.9 | NA |

**Table 5.** *Cont.*

| Reference | Multimodal Classifier | ADNI Subjects | Modalities | Performance Criteria (%) | | | | | | | |
|---|---|---|---|---|---|---|---|---|---|---|---|
| | | | | AD Vs. HC | | | | MCI Vs. HC | | | |
| | | | | AC | SE | SP | AUC | AC | SE | SP | AUC |
| [208] | Multitask feature selection method + MKL | 51AD + 99MCI + 52HC | MRI | 91.10 | 91.57 | 92.88 | 96.55 | 73.54 | 81.01 | 65.38 | 78.26 |
| | | | FDG−PET | 91.02 | 89.02 | 90.58 | 95.84 | 72.08 | 75.56 | 59.23 | 77.06 |
| | | | MRI + FDG−PET | 94.37 | 94.71 | 94.04 | 97.24 | 78.80 | 84.85 | 67.06 | 82.84 |
| [188] | Multivariate modeling + SVM | 50 MCI | MRI | NA | NA | NA | NA | 67 | 37 | 87 | NA |
| | | | FDG−PET | NA | NA | NA | NA | 62 | 10 | 97 | NA |
| | | | PIB−PET | NA | NA | NA | NA | 45 | 45 | 80 | NA |
| | | | MRI + PIB−PET | NA | NA | NA | NA | 76 | 53 | 90 | NA |
| | | | MRI + FDG−PET | NA | NA | NA | NA | 37 | 37 | 90 | NA |
| [184] | Multimodal biomarker classifier + SVM | 95AD + 182MCI + 111HC | MRI | 73 | NA | NA | 78 | 70 | NA | NA | 68 |
| | | | CSF | 82 | NA | NA | 85 | 74 | NA | NA | 77 |
| | | | MRI + CSF | 84 | NA | NA | 90 | 77 | NA | NA | 78 |
| | | | MRI + CSF + ApoE | 85 | NA | NA | 88 | 79 | NA | NA | 79 |
| [187] | M2TFS + MKL | 51AD + 99MCI + 52HC | MRI | 88.68 | 84.51 | 92.50 | 94 | 73.12 | 78.28 | 63.65 | 79 |
| | | | FDG−PET | 84.42 | 83.53 | 84.81 | 91 | 67.11 | 75.96 | 50.19 | 72 |
| | | | CSF | 82.26 | 82.55 | 81.54 | 87 | 70.72 | 71.62 | 69.04 | 75 |
| | | | MRI + FDG−PET | 95.00 | 94.90 | 95.00 | 97 | 79.27 | 85.86 | 66.54 | 82 |
| | | | MRI + FDG−PET + CSF | 95.40 | 94.71 | 95.77 | 98 | 82.99 | 89.39 | 70.77 | 84 |
| [186] | MKL with multitask feature learning | 51AD + 99MCI + 52HC | MRI | 92.25 | 92.16 | 92.12 | 96 | 73.84 | 77.27 | 66.92 | 77 |
| | | | FDG−PET | 91.65 | 92.94 | 90.19 | 96 | 74.34 | 85.35 | 53.46 | 78 |
| | | | MRI + FDG−PET | 95.95 | 95.10 | 96.54 | 97 | 80.26 | 84.95 | 70.77 | 81 |
| [209] | M3 | 16AD + 22HC | MRI + PET | 89.47 | 87.5 | 90.91 | NA | NA | NA | NA | NA |
| [210] | DNN | 180AD + 160 MCI + 204HC | MRI | 82.59 | 86.83 | 77.78 | NA | 71.98 | 49.52 | 84.31 | NA |
| | | 85AD + 67MCI + 77 HC | MRI + FDG−PET | 91.40 | 92.32 | 90.42 | NA | 82.10 | 60.00 | 92.32 | NA |
| [211] | CNN | 93 AD + 100 HC | MRI + PET | 89.64 | 87.1 | 92 | 94.45 | NA | NA | NA | NA |
| [17] | SVDD | 77 AD + 82 HC | MRI | 88.15 | 89.02 | 90.18 | 95.00 | NA | NA | NA | NA |
| | | | FDG−PET | 85.16 | 86.84 | 84.14 | 92.04 | NA | NA | NA | NA |
| | | | MRI + FDG−PET | 93.65 | 90.08 | 92.75 | 97.30 | NA | NA | NA | NA |

FDG: fluorodeoxyglucose, CSF: Cerebrospinal fluid, PIB: Pittsburgh compound B, ApoE: apolipoprotein E54 allel, HIS: Hue-intensity-saturation, MKL: Multi- kernel learning, SCLDA: Sparse composite linear discriminant analysis, M3T: Multimodal multitask, OPLS: Orthogonal partial least squares, M2TFS: Manifold regularized multitask feature learning, M3: Multimodal imaging and multi-level characteristics with multi-classifier, DNN: Deep neural network, CNN: Convolutional neural network, SVDD: Support vector data description.

3.2.1. Spatial Domain Techniques

- **Principal components analysis**: PCA used in many works [216–223] makes it possible to carry out a linear orthogonal transformation of a multivariate set of data which contains variables correlated with N dimensions in other containing new variables not correlated to M smaller size dimensions. The transformation parameters sought are obtained by minimizing the error covariance introduced by neglecting N-M of the transformed components. This technique is very simple and effective. It benefits from faster processing time with high spatial quality. It selects the optimal weighting coefficients according to the content of the information; it removes the redundancy present in the input image. It compresses a large amount of input data without much loss of information. However, a strong correlation between the input images and the merged image is necessary. In addition, the merged image quality is generally poor with spectral degradation and color distortion.

- **Hue-intensity-saturation:** HIS used in many works [224–228] converts a color image of the RGB space (red, green and blue) into an HIS color space. The intensity band (I) in the HIS space is replaced by a high-resolution panoramic image, then reconverted in the original RGB space at the same time as the previous hue band (H) and the saturation band (S), which creates an HIS fusion image. It is very simple, efficient in calculation and the processing time is faster. It provides high spatial quality and better visual effect. The change in intensity has little effect on the spectral information and is easy to manage. However, it suffers from artifact and noise which tend to weaken the contrast. It only processes multi-spectral bands and results in color distortion.

- **Brovery transformation:** It is a combination of arithmetic operations which normalize the spectral bands before they are multiplied by the panoramic image. It retains the corresponding spectral characteristic of each pixel and transforms all the luminance information into a high-resolution panoramic image. This technique proposed by many works [229–231] is very simple, effective on the computer level and has a faster processing time. It produces RGB images with a high degree of contrast. Good for multi-sensory images and provides a superior, high-resolution visual image. Generally the Bovery fusion image is used as a reference for comparison with other fusion techniques. However, it ignores the requirement for high-quality synthesis of spectral information and causes spectral distortion, which results in color distortion and high-contrast pixel values in the input image. It does not guarantee to have clear objects of all the images.

- **Guided filtering:** This technique [232–235] is based on a local linear model which takes into account the statistics of a region in the corresponding spatial neighborhood in the guide image while calculating the value of the output pixel. The process first uses a median filter to obtain the two-scale representations. Then the base and detail layers are merged using a weighted average method. It is very simple, in terms of computation and adaptable to real applications whose computational complexity is independent of the size of the filtering kernel. It has good smoothing properties preserving the edges and does not suffer from the gradient reversal artifacts observed when using a bilateral filter. It does not blur strong edges in the decomposition process. Despite the simplicity and effectiveness of this technique; however, the principal problem with the majority of guided filters is associated with ignorance of the structural inconsistency between the ground truth and target images such as color [234]. Moreover, the halos could represent an obstacle [235].

Several spatial domain techniques represent a simple means to obtain a fusion image but, due to degrading performance and weak or ineffective results, they have not been sufficiently applied, especially in real-time applications. Some of these methods are mentioned below.

- **Simple average:** The pixel value of each image is added. The sum is then divided by 2 to obtain the average. The average value is assigned to the corresponding pixel of the output image. The principle is repeated for all pixel values. This technique [217,236] is a simple way to obtain a fusion image with focusing of all the regions of the original images. However, the quality of the output

image is reduced by incorporating noise into the merged image, which results in undesirable effects, such as reduced contrast. In addition, the possibility of having clear objects from all of the images is not guaranteed.

- **Weighted average:** It calculates the sum of the pixels affected by coefficients, divided by the number of pixels. The weighted average value is assigned to each pixel of the input image to obtain the value of the corresponding pixel in the output image. This technique used in some works [217,231,237–239] improves the reliability of detection. However, there is a risk of increasing noise.

- **Simple block replacement:** In this technique [223,240], for each pixel, its neighboring pixels are added and a block average is calculated. The pixel of the merged image is obtained by taking the pixel with a maximum block average among all the corresponding pixels in the input image.

- **Max and Min pixel values:** These techniques are used in many works [217,236,240], they choose the focused regions of each input image by choosing the highest value (or the lowest in the case of the min pixel value technique) for each pixel. This value is assigned to the corresponding pixel in the merge image.

- **Max-Min:** For this technique [240], in the merged image, the output pixels are obtained by averaging the smallest and largest values of the corresponding pixels in all of the input images.

The last four techniques are easy to implement and provide several rules for merging images, most of which are very simple. However, they produce a fuzzy output that affects the contrast of the image, which limits their potential for real-time applications.

### 3.2.2. Frequency Domain Techniques

Discrete Transform

In signal processing, the discrete transformations represent in most cases linear transformations of signals between discrete domains, such as between discrete time and discrete frequency. In the case of medical signal processing, it provides a sparse representation of smooth images in pieces. The most used techniques based on this type of transformation are discrete cosine transform (DCT), curvelet transformation (CT) and wavelet transform.

- **Discrete Cosine Transform:** The DCT described in many works [241–246] makes it possible to perform a discrete transformation which provides a division into N × N pixel blocks by operating on each block. As a result, it generates N coefficients which are quantified to reduce their magnitude. It reduces complexity by breaking down images into series of waveforms. It can be used for real applications. However, the merged image is not of good quality if the block size is less than 8 × 8 or equivalent to the size of the image itself.

- **Curvelet Transform:** The CT used in many works [247–252] is a means of characterizing curved shapes in images. The concept is to segment the complete image into small overlapping tiles, then the ridgelet transformation is performed on each tile. The curvelet transform provides fairly clear edges because the curvelets are very anisotropic. They are also adjustable to properly represent or improve the edges on several scales. However curvelet do not have time invariance.

- **Wavelet Transform:** In wavelet-based fusion, used in different studies [217,220,253–258], once the image is decomposed by wavelet transformation, a multi-scale composite representation is constructed by selecting the salient wavelet coefficients. The most applied techniques based on wavelet transform are discrete wavelet transform (DWT), stationary wavelet transform (SWT) and Kekre's Wavelet transform (KWT).

    - **Discrete wavelet transform:** DWT [217,258] allows a discrete transformation for which the wavelets are discretely sampled. The key advantage of DWT is that it provides temporal resolution, in the sense that it captures frequency and location information. However DWT

requires a large storage space, the lack of directional selectivity and causes the loss of information on the edges due to the down-sampling process, effect of blurring, etc.

- **Stationary wavelet transform:** SWT [250,259], allowing a discrete transformation, begins by providing from the original image, information relating to the edges of levels 1 and 2. Then, a measurement of spatial frequency is used to merge the two contour images, and in order to obtain a complete contour image. From level 2 of the decomposition, the SWT offers satisfactory results. Its stationary property guarantees temporal invariance, which is obtained by suppressing the process of subsampling, but SWT is more complex in terms of computation and processing, which consume time. Moreover, although it performs better at separate discontinuities, its effectiveness degrades at the edges and textured locals.

- **Kekre's wavelet transform:** The KWT [260], which allows a discrete transformation to be carried out, is applicable for images of different sizes. Its results are generally good. In addition, different variations of KWT are simply generable, only by changing the size of the basic Kekre's transformation. However, this type of transformation is not explored enough. Intensive research is; therefore, desired to bring out its weaknesses.

- **Hybrid Approach-Based Fusion:** To achieve a merger, some researchers have used hybrid methods which allow two or more methods to be combined in a single scheme. Below some of them are presented.

    - **Hybrid SWT and curvelet transform:** With the hybrid SWT/CT technique [261] we first decompose the input images by applying the SWT in order to obtain the high- and low-frequency components. Thereafter, a curvelet transform is applied to merge the low-frequency components. The principle is based on the segmentation of the whole image into small superimposed mosaics, then for each mosaic, we apply the transformation into crest. Components with high frequencies are merged according to the largest coefficients of absolute value. The final fusion image is finally obtained using the inverse SWT transformation. This hybrid method makes it possible to avoid the drawbacks of the two combined methods, namely the block effects of the fusion algorithm applied by the wavelet transformation, as well as the performance defects of the details of the image in the curvelet transformation. It retains image details and profile information such as contours. It adapts to real applications. However, a lot of time is consumed.

    - **Discrete wavelet with Haar-based fusion:** In the DWT with Haar-based fusion [262], once the image is decomposed by wavelet transformation; a multi-scale composite representation is constructed by selecting the salient wavelet coefficients. The selection can be based on the choice of the maximum of the absolute values or of a maximum energy based on the surface. The final step is a transformation into inverse discrete wavelets on the composite wavelet representation. It provides a good quality merged image and better signal-to-noise ratio. It also minimizes spectral distortion. Different rules are applied to the low- and high-frequency parts of the signal. However, pixel-by-pixel analysis is not possible and it is not possible to merge images of different sizes. The final merged image has a lower spatial resolution.

    - **Kekre's hybrid wavelet transform**: This fusion technique [263] allowing a discrete transformation to be carried out, exploits various hybrid transformations to decompose the input images. Quoting the hybridization Kekre/Hadamard, Kekre/DCT, DCT/Hadamard, etc. Then the average is applied to merge the decomposed images, and to obtain its transformation components. The latter are subsequently converted to an output image by applying a reverse transformation. Like its similar KWT, its advantage is that it is applicable for images that are not only a whole power of 2.

    - **Hybrid DWT/PCA:** The DWT/PCA fusion [220] allows for a discrete transformation. It provides multi-level merging where the image is double-merged, which provides an

improved output image containing both high spatial resolution and high-quality spectral content. However, this kind of merger is quite complex to achieve.

Pyramid Decomposition

An image pyramid consists of a set of low-pass or band-pass copies of an image, each copy representing pattern information of a different scale. In a picture pyramid, each level corresponds to a factor two smaller than its predecessor, and the highest levels will focus on the low-spatial frequencies. An image pyramid contains all the information necessary to reconstruct the original image. Among the techniques based on pyramidal decomposition, we could cite the Laplacian technique [250,264–270], Gaussian technique [269] gradient pyramid [270], low-pass ratio pyramid [271], contrast [272] and the morphological technique [273]. Pyramid techniques provide good visual quality for a multi-focus image. However, all pyramid decomposition techniques produce more or less similar results. In addition, the number of decomposition levels affects the result of the merge.

### 3.3. Critical Discussion about Multimodal Fusion Techniques

Various multimodal fusion techniques have been exploited by several works related to brain imaging. We could break these techniques down into two types: Those that can be used in the frequency domain where the Fourier transform of the input image is first calculated, then the inverse Fourier transform is determined to provide the output image. Other fusion methods are of the spatial domain which are interested in the pixels of the input images whose modifications are made on the values of the pixels to provide the desired output image.

In general, researchers have preferred to use pixel level fusion methods that have provided the best results such as IHS, PCA, independent component analysis. (ICA), guided filtering and Brovey transformation, etc. However, these techniques suffer from spectral degradation. Techniques using traditional fusion rules have also been used such as weighted average, absolute maximum, etc. Unfortunately, they provide a poor quality output image because the information from the low- and high-frequency coefficients is overlooked or used inefficiently. Since medical images are sensitive to the human visual system which exists on different scales, this type of technique is undesirable for performing the fusion.

Pyramidal decomposition [250,264–274] and multi-resolution [275,276] techniques have solved this problem, citing fusion by the gradient pyramid, the Laplacian pyramid, the contrast pyramid, etc. On the other hand, the output image suffers from a reduced contrast with pyramidal fusion and slightly less with multi-resolution techniques. In addition, these techniques produce blocking effects, because in their decomposition process, no spatial orientation selectivity is taken into account.

With the progression of multi-resolution fusion, wavelet (WT) transformation was widely used [205,217,220,250,253,254,256–258] to merge medical images, particularly DWT, since it maintains spectral information. However, the spatial characteristics are poorly expressed. Added to this, the isotropic WT is devoid of shift invariance and multi-directionality. It also does not provide optimal expression of highly anisotropic edges and contours in brain images [183]. In the output image, it is difficult to surely preserve all of the salient features of the input images. This eventually causes an inconsistency in the results of the merger, and introduces artifacts. In addition, WT offers efficient fusion only for isolated discontinuities. Unfortunately, its performance deteriorates on edges and textured regions. On the other hand, it reserves limited directional information along the vertical, horizontal and diagonal directions [183].

Multi-scale geometric analysis (MSGA) [193] has overcome these limitations, thanks to its proposed techniques which allow multi-scale decomposition for high dimensional signals. Citing the ridgelet, curvelet, bandlet, brushlet and contourlet techniques [247–252,276]. The contourlet technique [193,250,276] is a lean 2D representation for 2D signals. It allows better capture of 2D geometric structures in visual information than traditional multiscale methods. In addition, the extended version, the non-subsampled contourlet transform (NSCT) [183,191,192,276–280], inherits all

the advantages of contourlet transformation by adding the characteristic of invariant decomposition by offset, which effectively suppresses pseudo-Gibbs phenomena. This increases performance thanks to the use of directive contrast which takes advantage of contrast and visibility. It also improves the quality of the output image, especially around the edges [183], by producing a more noticeable and more natural merged image. Some works have attempted to propose new approaches which go beyond the conventional context of fusion such as neural networks with pulse-coupled neural network (PCNN) [191,281], fuzzy logic [282], genetic algorithms (GA) [283] and independent component analysis (ICA) [284].

Recent trends to apply hybrid fusion techniques like hybridization weighted average/Brovery [231], HIS/PCA [285,286], Laplacian/maximum likelihood [265], IHS/WT [287], contourlet/PCA [288], Laplacian/DCT [266], Bloc replacement/PCA [223], NSCT/PCNN [191], WT/PCNN [281], Laplacian/histogram equalization [267], GA/WT [289], DTW/PCA [290], etc. The objective is to increase the performance of conventional medical image fusion systems, and to meet the needs of real medical applications. For example image fusion using the IHS/WT hybrid transform [287] improves the synthetic quality of the merged image. The fusion by IHS improves the textural characteristics in the merged image, and the spatial details of the multi-spectral image. However, the spectral distortion is severe in the output image. WT could remedy this problem as it provides high-quality spectral content. In addition, hybridizing DWT with PCA [290] or other space domain methods improves performance compared to using the techniques separately.

In conclusion, traditional fusion techniques [291] suffer from several drawbacks and do not meet the requirements of current medical applications. The reason why, researchers have started a new research by focusing on different tracks in order to increase the performance of CAD systems. Among the addressed tracks, the tendency to apply hybrid models which combine the advantages of several conventional fusion techniques. These models could be the future trend for neuroimaging research.

### 3.4. Critical Discussion about the Multimodal Diagnosis of AD

Several CAD systems [7,14,144,150–153,156] have been implemented over the past decade for the diagnosis of AD or its early-stage MCI. To distinguish patients with AD from those in normal aging, the researchers used machine learning methods such as SVM, ANN and naive Bayes classifier. However, it should be noted that these techniques have been used for single-modal images (MRI or PET) in the majority of the work. Unfortunately, few works have exploited multimodality for the diagnosis of AD like [10,17,46,154,184,186,187,206,207], which seems to be better achieved by serving the advantages of several types of images, each measuring a different type of structural or functional characteristic. In reality, the various imaging environments and modalities offer complementary information that is useful when used in conjunction. This improves the performance of the diagnostic system compared to the system using a single modality.

In this context, some research groups have adapted machine learning algorithms to several types of modalities (or to additional clinical/cognitive data), by concatenating the structural and functional characteristics of each subject into a single vector of characteristics. However, this type of approach causes strong growth in the distribution dimension. In addition, the principle prompts us to find the right standardization for each modality in order to preserve its informational content and avoid overwhelming the characteristics derived from one or the other type of image. The difficulty in this case is in the evaluation of the relevance of the content of each modality, which allows the optimization of the classifier.

To remedy this problem, several researchers [186,187,203,204,208] have exploited the idea of carrying out multi-kernel learning (MKL) based on the combination of several kernels. This learning applied to MRI/PET multimodal images is more flexible, thanks to the use of different weights on the modality biomarkers. In addition, it provides a unified means for combining heterogeneous data when a different data type cannot be directly concatenated. However, despite its performance in terms of

cross-validation, this feature selection method requires having the same number of features calculated for each modality.

In addition to the use of MRI/PET multimodal neuroimaging data, studies have attempted to integrate other types of data from other biomarkers and test these prognostic capacities such as CSF, Pittsburgh compound B (PIB)-PET, apolipoprotein E (ApoE) and genetic data. The advantage of these measures developed for the diagnosis of AD is that the number of its characteristics is different, and that they often contain relevant and complementary pathological information which can help in the future diagnosis of AD and increase the performance of the classification [292].

For example, researchers, as in [10,206], proposed a kernel-based combination to integrate the CSF with the FDG-PET and MRI modalities. They have shown that the morphometric changes in AD and MCI are linked to the CSF, which offers additional information [293]. Additionally, the [11C] PIB-PET biomarker has been used by several longitudinal studies and therapeutic interventions to estimate the evolution of AD. The PET imaging tracer, PIB, was developed to identify the cerebral amyloid. However, quantitative analysis of PIB [11C] data requires the definition of regional interest volumes. In this context, to define the regions for a PIB-PET analysis, researchers as in [188,294] have shown that the integration of MRI or PET offers similar results. This avoids the need for an MRI that takes time and increases costs. Therefore, MRI analysis remains more appropriate for clinical research, while the application of a PET model to [11C] PIB is adequate for clinical diagnosis. Studies have been interested in the application of the ApoE4 genotype that has been integrated into the MRI and CSF in several works [184,185], which have tested the performance of the classification after stratification of ApoE4. This biomarker was commonly applied as a stratification factor or covariate that contributes to the adjustment of heterogeneity in sporadic AD, since non-e4 status in EOAD patients is correlated with typicity.

Biological or genetic biomarkers [184,185] have also been developed for the diagnosis of AD or its early-stage MCI. Studies by genetic endophenotype by imaging would make it possible to develop the link between genetics and the topography of AD by focusing on the areas of the brain most associated with pathological genotypes. However, the use of these biomarkers differs from one research center to another and is subject to various factors, such as the cost of local availability and historical patterns of use [207]. In general, several deductions have been noted by researchers from these centers, namely: (1) The increase in total tubulin-associated unit (t-tau) proteins and hyperphosphorylated tau at the level of threonine 181 (p-tau) in the CSF is associated with the pathology of neurofibrillary tangling; (2) the decrease in the amyloid β (Aβ42) is indicative of the pathology of the amyloid plaque; and (3) the presence of the $\varepsilon$4 allele of APOE can predict cognitive decline or conversion to AD [184]. However, the change in the proteins t-tau and p-tau in the CSF does not really express the extent of the tau pathology in the Alzheimer's brain. These CSF biomarkers are only peripheral substitutes for the current pathology of the tau protein. Recently, several ligands of the tau protein are being developed. Such technology offers a much more precise measure of the pathology of the tau protein, which significantly promotes better understanding and classification of the early and presymptomatic stages of AD.

It should be added that most large-scale works have exploited multimodal techniques which target the exploration of the relationship between various modalities for the same participants, neglecting the useful relationship between the different participants. In another sense, the techniques for selecting the multitasking functionalities are used only to jointly select the common functionalities through different modalities, ignoring the information relating to the distribution of the data in each of these modalities, which is important for a later classification. Some researchers as in [186,187] have tried to tackle this problem, by proposing a method for learning multiple regularized multitasking functionalities, which makes it possible to preserve both the inherent relationship between the data of the different modalities and the information relating to the distribution of data in each modality.

However, we can generally suggest a few remarks found in most of the works cited in the literature:

- All the classification models performed very well, distinguishing participants with normal aging from the patients with AD. The performance was worse for MCI vs. AD determination, which proved more difficult, probably due to the MCI biomarker scheme which is quite similar to that observed in AD. However, multimodal classification had better diagnostic and forecasting power than single-modal classification.
- The majority of clinical studies in recognized biomedical laboratories have focused on binary classification problems (i.e., AD vs. HC and MCI vs. HC), neglecting to test the power of the models proposed for multi-class classification of AD, MCI and normal controls. It is true that the latter type of classification is more difficult to verify than the binary classification, but it is crucial to diagnose the different stages of dementia. In addition, longitudinal data may contain essential information for classification, while the proposed studies can only deal with basic data.
- Few studies have used two or more biomarkers simultaneously among MRI, PET and CSF for the diagnosis of AD, using images from synthetic bases such as Alzheimer's Disease Neuroimaging Initiative (ADNI) [293] and Open Access Series of Imaging Studies (OASIS) [295] to combine MRI with other modalities applying image fusion methods. The ADNI database, launched in 2003 by various institutes, companies and non-profit organizations, represents one of the largest databases to date; however, the limit of the ADNI data set is that it is not neuropathologically confirmed, which is like even a fairly delicate task to perform in practice.

## 4. Conclusions and General Requirements

Many advanced countries with a long life expectancy, such as Canada, Japan and the United States, are moving toward an aging society, and thus the number of patients with brain diseases, including some dementia disorders, will increase as AD with the rise in average life expectancy. Neuroradiologists expect that the CAD systems can assist them in diagnosing brain diseases by providing useful information. Therefore, the CAD systems for brain diseases, especially for AD which is the most common form of dementia, will become more essential for neuroradiologists in clinical practice in the near future. Moreover, because MR neuroimaging used in practice is one of the most widely used imaging modalities to establish trusted clinical settings in brain medical studies in this regard, this review provides a preliminary summary and reported some studies of researchers which attempt to develop very useful magnetic resonance CAD systems for brain disorders and AD in particular. In this context, the various machine learning techniques that have been explored for in vivo classification, and particularly segmentation phase in the CAD MR brain system, have been described and criticized, specifying the disadvantages and advantages of each of them. We are interested in machine learning methodologies because they are highly flexible to the inclusion of expert knowledge and have been demonstrated in numerous applications to perform accurately and robustly.

In the context of the brain regions segmentation, despite intensive research in this area, there is currently no reliable method for all types of images. Nevertheless, a set of ad hoc methods that has received a certain degree of popularity and that we have divided into five categories are: Techniques based on the shape approach, based on the contour approach, based on the region approach, based on theory graphs and those based on the structural approach.

In the context of the classification process, existing work reported in the literature has shown that the classification of brain images is possible via techniques based on supervised or unsupervised learning. However, due to the specificity of the brain images in which the CAD system must be formed, based on the field of truth or clinical evidence, the researchers preferred to apply methods based on supervised learning and rejected unsupervised classification, which does not take into account training data.

However, because of the drawbacks raised by the classification and segmentation methods, it has been found that different researchers have turned to new tracks to maximize the accuracy of CAD systems. Among these tracks, the hybrid models which make it possible to combine the advantages of various techniques derived from "soft computing".

MRI provides high-resolution images with anatomical information. On the other hand, functional images, such as PET, SPECT, etc., provide low-spatial resolution images with functional information. Therefore, a single modality of medical images cannot provide comprehensive and accurate information. As a result, combining anatomical and functional medical images to provide much more useful information through image fusion, has become the focus of imaging research and processing. Different biomarkers provide complementary information, which have been shown in the literature to be useful in neuroimaging and AD diagnoses when used together. Thus we showed in this survey paper that the combination of MRI and other modalities using data mining methods, as a tool, more accurately classifies brains disorder patients as AD subjects at the baseline compared to using either biomarker separately. For this purpose, some results of related-work were reported, and the most used image fusion methods were summarized by specifying the disadvantages and the advantages of each one.

Several multimodal fusion methods have been devised by the researchers. We divided them into two types: Those which can be exploited in the frequency domain and others used in the spatial domain where the pixels of the input images undergo improvements. In reality, the latter type of technique was preferred in many studies.

However, the conventional methods applied for multimodal fusion suffer from several drawbacks and do not meet the requirements of current medical applications. In this context too, hybrid models combining the advantages of conventional techniques have been applied for the purpose of improving the performance of CAD systems based on multimodal fusion.

We noticed that the efficiency of the CAD systems proposed by researchers is demonstrated by estimating the percentage of several performance measures, such as sensitivity (SE), which represents the true positive rate; specificity (SP), which estimates the true negative rate; and precision (AC), which determines the proportion of true results in the database, whether true positive or true negative.

The authors tried to prove that the proposed methods performed fairly well compared to a selection of other methods. Indeed, we have observed that certain techniques have obvious advantages over others, based on the speed of processing time, or on the lesser complexity or even on the lower memory requirements. However, it is not possible to draw absolute conclusions about what is best or worst without first performing in-depth tests on all the proposed methods.

Furthermore, a potential limitation of combining biomarkers is limited by the practical clinical implications as imposed by the medical experts based on the requirements of specific medical studies. In addition to medical reasons, there exists technical challenges in image fusion resulting from image noise, resolution difference between images, inter-image variability between the images, lack of sufficient number of images per modality, high cost of imaging and increased computational complexity with increasing image space and time resolution. Many of these challenges remain open and the problem is much more significant in developing fusion image algorithms for real-time medical applications such as robotic-guided surgery. Nonetheless, even under these challenging situations, the fused images provide the human observers improved viewing and interpretation of medical images; and the image fusion in multimodal medical imaging environment has proved to be useful and the trust in its techniques is on the rise.

It is expected that the innovation and practical advancements would continue to grow in the upcoming years. In this context, some general requirements on the application of multimodality fusion that emerge from this survey, especially for fusion algorithms, could be suggested as follows: (1) The algorithm should be able to extract and integrate complimentary features from the input images; (2) it must not introduce artifacts or inconsistencies according to the human visual system; (3) it should be robust and reliable. Generally, these can be evaluated subjectively or objectively. The former relies on human visual characteristics and the specialized knowledge of the observer, hence are vague and time-consuming but typically accurate if performed correctly. The other one is relatively formal and is easily realized by the computer algorithms, which generally evaluate the similarity between the fused and source images. However, selecting a consistent criterion with the subjective assessment of the

image quality is rigorous. Hence, there is a need to establish an evaluation system and an evaluation index system is set up to evaluate the proposed fusion algorithm.

**Author Contributions:** Conceptualization, L.L.; methodology, L.L.; formal analysis, L.L.; investigation, L.L.; resources, L.L.; data curation, L.L.; writing—original draft preparation, L.L.; writing—review and editing, L.L.; visualization, L.L.; Supervision, M.B.; Funding acquisition, L.L., M.B. and O.A.M. All authors have read and agreed to the published version of the manuscript.

**Funding:** This research was funded by the following organizations: FRQNT (Fonds de recherche du Québec—Nature et technologies), Quebec, Canada; ReSMiQ (Regroupement Stratégique en Microsystèmes du Québec), Quebec, Canada; L'Oréal-UNESCO for Women in Science, Paris, France and Caisse Desjardins de Sault-au-Récollet, Quebec, Canada.

**Acknowledgments:** The authors thank the following organizations: FRQNT, ReSMiQ, L'Oréal-UNESCO and Caisse Desjardins de Sault-au-Récollet for their financial support offered to accomplish this research. Special thanks to the evaluators of this work for the relevant comments.

**Conflicts of Interest:** The authors declare no conflict of interest.

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
