# Peer review of "A Survey on Computer-Aided Diagnosis of Brain Disorders through MRI Based on Machine Learning and Data Mining Methodologies with an Emphasis on Alzheimer Disease Diagnosis and the Contribution of the Multimodal Fusion"

_applsci, doi:10.3390/app10051894_

Round 1

Reviewer 1 Report

The authors present a review of several diagnosis (CAD) techniques of brain disorders based on MRI data and that use machine learning solutions to the problems of segmentation of the brain and classification of the brain's condition. They focus the scope of their review to publications addressing the diagnosis of Alzheimer disease. Finally, they differentiate methods combining several imaging modalities to the ones that only rely on MRI data.

In general, the authors made an good job joining works from as far as 1996 and up to 2020 (ref CHI20). I believe the readability could be improved as well as the scope and objectives of the review paper. Below I am writing all the suggestions I would ask of the authors.

Major comments

English writing: The English writing is overall very good, although I did find some typos so run the manuscript through a spell check before resubmitting. Also, as I mention below, consider including more context, extended captions and joining statements between the sections to improve readability.

Improve readability. Overall, the authors present a lot of information with a clear effort of synthesising the information. I think the paper lacks on readability, and it seems more of a collection of figures and table with not enough context. A few lines in between sections, to drive the reader, together with improving the captions of the tables and figures would be very helpful.

1. Abstract: The abstract should present a snapshot view of the paper’s contribution, including the context, data utilised, methods and findings. A somewhat vague representation of the paper’s findings can be read from the abstract. Some of the information is already there, therefore I suggest a thorough rewording of the abstract, including some results and major contributions.

2. Introduction. I think the author’s concise introduction and structure is very good. The objective is clear, although I believe that given the amount of literature reviewed, I would encourage the authors to extend the scope of their work to more than “summarising” the applications. I believe a more thorough discussion (and hence, conclusion) could be reached with a bit more analysis of the measuring parameters and comparisons.

3. Presentation of the techniques. I think the literature review is thorough and well synthesised. Some of the figures and tables could be improved and, as mentioned before, the readability should be improved. Enhance each table’s caption with noteworthy aspects that could be read. These would also improve the Sections called “Critical Discussion”.

3. Discussion and Conclusion. Give a more comprehensive conclusion to your work by combining the points made in the discussion, and how the research objectives were met. I think the authors could have delved deeper into the individual results of the different articles and the sensitivity of the different techniques. For example, I could not find the authors’ notes on false positives or false negatives, or sensitivity to errors. Segmentation techniques could be varying in the order of pixels, and even then, there is an error margin that should be considered and discussed (especially if dealing with something that’s looking to become widely used).

Other comments.

- Figures and tables should be self-contained, meaning that a reader should be able to understand a figure and its caption without needing to go search for an explanation in the text. I suggest the authors expand on this issue for all Tables and figures presented. The inclusion of the legend at the end of the figure or table is appreciated, however a small extension would improve the caption much more.

- Have consistency with the design, font and sizes all the Tables and try to keep some overall design rule over the Figures. Some tables have a significantly larger font than others.

Particular observations:

Figure 1. Consider linking this figure to the rest of the paper by referring to the sections each block will correspond to.

Figure 2. A proposed better caption would be: “Classification of various MR brain image segmentation methods described in the literature. In this work, the methods were classified in five categories based on the approach (green): Form, Structural approaches, Graph theory, Region and Contour.”

Table 1. Is there anything the authors would like to address regarding the Measuring parameters and comparison methods? Did the authors find any patterns? The methodologies in both columns seem very different from each other. I wonder if this could be clarified with a more thorough caption explaining the columns, or whether the authors could discuss on these methods.

Table 2. It would be helpful to have some sort of link from the diagram in Fig 2 to Table 2 or into Sections 2.2.1.X. It also seems to be too general, given Sections 2.2.1.X, maybe the authors could make several tables with the key algorithms per category?

Figure 3. Same as Figure 2

Table 3. Similar to table 2 in terms of referencing the diagrams to the tables and to the text (Sections 3.1.2.X). Consider expanding the captions and including any comments on the performance measures. Regarding the Computation time column, only two techniques have an input, but they are not related to each other. [KLO08] reports training time while [ZHA11a] reports computation time for one image. I think these could be mentioned in the caption and be taken from the table, to give more space for the rest of the columns.

Figure 4. Same as Figures 2 and 3

Reviewer 2 Report

  The authors have written a very comprehensive review of the techniques that can be used in all aspects of computer aided diagnosis of Alzheimers.  The techniques covered have ranged from segmentation of the MRI images to specific biomarkers of the disease.  Even within a specific technique, the authors have been thorough to make the reader aware of the tremendous amount of methods available to accomplish each task.  The authors remain impartial throughout, presenting the pros and cons of each, with no editorials showing a preference for a particular technique over others. 

 We feel the authors have written a thorough overview of the various techniques used for segmentation of the brain.  We appreciate going into the pros and cons of the many various algoirithms present in the literature.  We feel this portion of the review could be enhanced by also highlighting some of the existing tools as opposed to simply referencing papers and techniques.  If a small section were dedicated to professionally developed software such as the FMRIB Software library or the Insight Segmentation and Registration Toolkit, which comes packaged with many of the referenced segmentation algorithms, we feel the section would be enhanced.  It also would provide a baseline upon which researchers could begin applying some of the techniques more specialized to Alzheimers diagnosis.

Possible Typo:  In line 470-471 under the “Critical Discussion About Segmentation Techniques” section, there is a sentence:  “Easily adaptable with neural networks, MRF and histogram thresholding.”  We feel this is a typo, and encourage the authors to thoroughly review the manuscript and revise any other typos before publication.

            On lines 479-480, the authors state that it is “beneficial to combine all or most of the measures for better segmentation results”.  While we agree with the authors, we feel that this statement should be justified by explicit examples and references, so as to strengthen the argument.

While we appreciate the thoroughness with which the authors have left nearly no widespread techniques out of the review, we do feel that it stands out when some techniques are given several paragraphs of analysis, while others receive only a few sentences.  This is particularly evident in section 3.2 on multimodal fusion techniques.  We suggest that the authors consider moving the less analyzed techniques to the end, and collect all such techniques in a paragraph.  In this way, the reader will be made aware and can easily identify the most widespread and successful algorithms , while also receiving some information about some of the less applied algorithms.

Possible Typo:  In line 688, under the Hue-intensity-saturation, the passage “However, she suffers from artifact...”  appears to by a typo.  The authors also claim this tends to increase the contrast, and present this as a negative. 

Possible Typo:  In line 705, under guided filtering, the statement “The process first uses a medium filter…”   We feel this should be median filter.    

The authors have presented a strong case for the Guided Filtering technique.  However, especially compared to other techniques covered, there is a noticeable lack of negatives of this specific technique presented in the review.  If the authors could cover a few of the flaws, the technique could be more accurately compared to the other methods. 

Possible Typo:  In line 769 under the stationary wavelet, the authors claim “However, it consumes enough time.”   We feel if the authors were more explicit with the time duration and how that limits the appeal of the method, the section would be improved. 

The authors cite reference FAL10 many times throughout the review.  While it does provide an overview of several techniques, it is also only a 6 page conference paper, and does not provide a significant amount of depth in any technique.  We suggest the authors consider using other references which go more in depth into the specific technique described, as opposed to other review-like references.  This way a reader interested in a specific technique can be made immediately aware of the standard references for that technique present in the literature.   
